# Inpainting the Neural Picture: Inferring Unrecorded Brain Area Dynamics from Multi-Animal Datasets

**Ji Xia**
Center for Theoretical Neuroscience,
Zuckerman Mind Brain Behavior Institute,
Kavli Institute for Brain Science,
Columbia University
jx2484@columbia.edu

**Yizi Zhang**
Department of Statistics,
Center for Theoretical Neuroscience,
Zuckerman Mind Brain Behavior Institute,
Kavli Institute for Brain Science,
Grossman Center for the Statistics of Mind,
Columbia University

**Shuqi Wang**
Laboratory of Computational Neuroscience,
École polytechnique fédérale de Lausanne (EPFL)

**Genevera I. Allen**
Department of Statistics,
Irving Center for Cancer Dynamics,
Center for Theoretical Neuroscience,
Zuckerman Mind Brain Behavior Institute,
Kavli Institute for Brain Science,
Columbia University

**Liam Paninski**
Department of Statistics,
Center for Theoretical Neuroscience,
Zuckerman Mind Brain Behavior Institute,
Kavli Institute for Brain Science,
Grossman Center for the Statistics of Mind,
Columbia University

**Cole Lincoln Hurwitz**
Center for Theoretical Neuroscience,
Zuckerman Mind Brain Behavior Institute,
Kavli Institute for Brain Science,
Grossman Center for the Statistics of Mind,
Columbia University

**Kenneth D. Miller**
Center for Theoretical Neuroscience,
Zuckerman Mind Brain Behavior Institute,
Kavli Institute for Brain Science,
Columbia University

## Abstract

Characterizing interactions between brain areas is a fundamental goal of systems neuroscience. While such analyses are possible when areas are recorded simultaneously, it is rare to observe all combinations of areas of interest within a single animal or recording session. How can we leverage multi-animal datasets to better understand multi-area interactions? Building on recent progress in large-scale, multi-animal models, we introduce **NeuroPaint**, a masked autoencoding approach for inferring the dynamics of unrecorded brain areas. By training across animals with overlapping subsets of recorded areas, NeuroPaint learns to reconstruct activity in missing areas based on shared structure across individuals. We train and evaluate our approach on synthetic data and two multi-animal, multi-area Neuropixels datasets. Our results demonstrate that models trained across animals with partial observations can successfully in-paint the dynamics of unrecorded areas, enabling

39th Conference on Neural Information Processing Systems (NeurIPS 2025).

multi-area analyses that transcend the limitations of any single experiment. Code is available at the following github repository: NeuroPaint

# 1   Introduction

Understanding how brain areas coordinate their activity to support complex behaviors, such as memory-guided actions and perceptual decision-making, is a central challenge in systems neuroscience [17]. Advances in electrophysiological recording techniques now enable single-cell, single-spike resolution measurements across dozens of interconnected brain areas [25, 43, 44]. These developments have driven the collection of brain-wide datasets from hundreds of behaving mice, creating new opportunities to study distributed neural computation at scale [28, 10, 27, 30]. Capitalizing on these datasets requires computational methods capable of modeling inter-area interactions across animals and tasks.

Traditional approaches to modeling inter-area interactions rely on pairwise neuronal correlations [12] or low-dimensional linear methods such as communication subspace analysis [39]. More expressive models, such as DLAG [15], mDLAG [16], and dCSFA [22], capture aspects of temporal and spatial dependencies but typically assume a fixed communication structure and can scale poorly to large datasets. Most recently, large-scale, multi-animal approaches like multi-task-masking (MtM) have shown that pretraining across animals with recordings from overlapping brain areas can improve cross-area prediction [61]. However, these methods are limited to modeling only the areas observed within a given recording session and do not take advantage of a key opportunity in multi-animal datasets: using data from multiple recording sessions across animals to infer activity in unrecorded areas of any given session.

In this paper, we introduce a new framework for analyzing multi-animal, multi-area recordings, NeuroPaint, that explicitly leverages shared anatomical structure across animals to model both recorded and unrecorded brain areas. Building on the masked transformer architecture introduced in [61], our approach incorporates two key innovations: (1) a masking strategy tailored for unrecorded areas, in which unrecorded brain areas are treated the same as masked areas during inference; (2) an architecture that models latent dynamics separately for each brain area, enabling interpretation of area-to-area interactions. To our knowledge, this is the first method specifically designed to infer the dynamics of unrecorded brain areas using data from both other animals and other sessions of the same animal.

We evaluate our method on synthetic data and two large-scale, brain-wide mouse datasets [9, 28]. We find that training across animals with overlapping subsets of recorded brain areas enables NeuroPaint to reliably infer the latent dynamics of missing areas. We compare our approach to LFADS [46] and generalized linear regression, in which we directly predict activity in missing areas from observed activity, demonstrating that our inferred latents provide significantly more predictive information about unrecorded areas. Taken together, this work introduces a new paradigm for analyzing brain-wide activity, demonstrating how shared structure across animals can be exploited to in-paint dynamics from missing brain areas and opening new possibilities for multi-area analyses that transcend the limitations of single-subject recordings.

# 2   Related work

**Multi-area models.**   Advances in electrophysiology now enable simultaneous recordings across multiple brain areas [25, 44, 59, 50]. To study concurrent signaling among distributed neuronal populations [1, 24, 28], generative models such as DLAG and mDLAG have been developed to uncover shared latent dynamics across brain areas [22, 15, 16]. While these models offer interpretability, they typically assume fixed bidirectional communication structures and suffer from limited scalability. Multi-area recurrent dynamical models have also been proposed to capture inter-area interactions more flexibly [20, 35, 34, 14]. However, none of these approaches can infer the dynamics of brain areas that are unrecorded during a given session.

**Stitching or quilting multi-session recordings.**   Recently, several groups have developed so-called "stitching" [42] or "quilting" [53] approaches to infer functional connectivity or latent dynamics from multi-session recordings with partially overlapping sets of neurons. Some of these methods

directly estimate the covariance [5, 6, 63], noise correlations [41, 55], or functional connectivity using generalized linear models [42] and graphical models [53, 7], while others infer latent factors or neural dynamics across sessions [51, 31]. Among these, LFADS [33] is most relevant to our work, as it infers latent dynamics using a nonlinear state space model and can be extended to multi-session settings via linear stitchers. However, state space approaches process time points sequentially and impose strong assumptions on temporal dynamics and noise structure, limiting their scalability and flexibility in large-scale, multi-animal datasets. Moreover, these approaches cannot directly infer neural dynamics in unrecorded brain areas.

**Large-scale models for neural analysis.** Recent work suggests that scaling models across animals and brain areas can be beneficial, as neural activity exhibits shared structure across individuals and regions [37, 2, 58, 60–62]. Transformer-based models have emerged as powerful tools for modeling these multi-animal neural datasets, including methods such as POYO+ [3], a multi-task decoder across animals; NDT [58], which uses masked modeling for neural prediction; and MtM [61] and NEDS [62], which utilize multi-task-masking to learn spatiotemporal structure in neural activity and the bidirectional relationship between neural activity and behavior. However, these models overlook a key opportunity in multi-animal, multi-area datasets: inferring neural dynamics in unrecorded brain areas of one animal by leveraging shared structure across animals.

# 3 Methods

In this work, we present **NeuroPaint**, a transformer-based masked modeling approach that predicts neural dynamics in both recorded and unrecorded brain areas using observed activity. In multi-animal extracellular recordings, each session samples neurons from only a subset of brain areas, leaving others unrecorded (Fig. 1A, 3A, E). With sufficient overlap across animals, shared structure can be leveraged to infer missing activity. To model this, we randomly mask recorded areas and train the model to reconstruct them, using a set of low-dimensional latents for each brain area (even when unrecorded), to enable inference of missing brain area dynamics across sessions.

## 3.1 Architecture

The architecture of NeuroPaint has four main components (Fig. 1C): (1) a cross-attention [23] stitcher that maps neural activity from each unmasked brain area to area-specific embedding factors; (2) a tokenizer that transforms these embedding factors into tokens and adds mask tokens for masked and unrecorded areas; (3) a transformer encoder that processes all tokens to produce latent factors for each brain area; and (4) a generalized linear stitcher that reconstructs neural activity from latent factors via a linear readout followed by an exponential nonlinearity. Most parameters are shared across sessions, with a few session-specific parameters noted below.

**Cross-attention read-in stitcher.** We use a cross-attention stitcher (shown in Fig. 1B) to map neural activity into a shared latent space, producing area-specific "embedding factors" that are consistent across sessions. This module builds on the Perceiver-IO architecture [23, 2], using latent tokens to reduce input length. Neural activity is tokenized at the neuron level, with each neuron providing keys and values. A fixed set of learnable latent tokens act as queries, ensuring that the output embedding factors have the same size as the latent tokens. To construct each neuron token, we concatenate the neural activity embedding with three learnable embeddings: an area embedding (encoding the neuron's brain area), a hemisphere embedding (indicating left or right hemisphere), and a unit embedding (unique to each neuron). The cross-attention stitcher then outputs a distinct set of embedding factors for each brain area. For a more detailed description of the cross-attention stitcher, see Appendix A.2.

We adopt a cross-attention stitcher [2, 3] in place of a linear stitcher [33, 61, 57], as the input-side transformation must be highly expressive. Neural activity can vary substantially across sessions, and aligning it into consistent across-session dynamics requires non-linear transformations. At the same time, minimizing the number of parameters is critical to avoid overfitting. The cross-attention stitcher supports parameter sharing across sessions and brain areas, improving generalization. Only the unit embeddings remain session-specific, as each session includes a distinct set of recorded neurons.

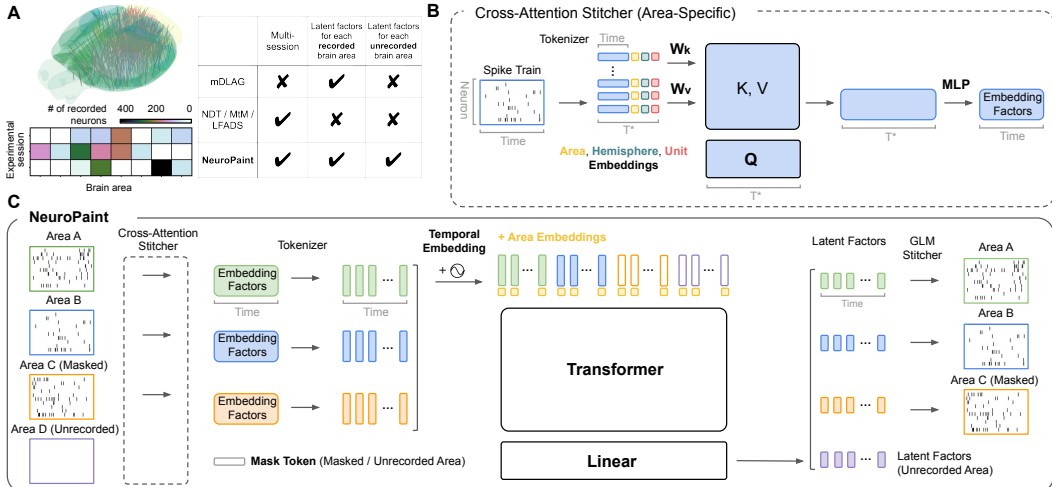

Figure 1: *Schematic illustration of NeuroPaint.* **(A)** Neuropixels probes only record a subset of brain areas simultaneously. The brain schematic summarizes probe locations across IBL sessions, and the table below shows variability in recorded neuron counts by area and session. In contrast to previous approaches (e.g., mDLAG [16], NDT [57, 58], MtM [61], LFADS [33]), NeuroPaint is the first method that can infer latent dynamics for each brain area, including unrecorded areas, using large-scale multi-session, multi-animal datasets. **(B)** We use a cross-attention stitcher to convert spike data into tokens, and concatenate area, hemisphere, and unit embeddings to form the keys and values. We use learnable latent tokens as queries to reduce input length and produce area-specific embedding factors. Bold symbols indicate parameters shared across sessions, while non-bold symbols denote area-specific parameters. **(C)** NeuroPaint uses a transformer-based architecture with cross-attention stitchers (as shown in B) to encode spike counts into area-specific embedding factors that are then tokenized into temporal tokens. We add temporal and area embeddings to the tokens, which are passed through the transformer and a linear layer to produce area-specific latent factors. During training, we mask tokens from sampled brain areas, and predict them using area-specific GLM stitchers.

**Tokenizer for embedding factors.** The cross-attention stitcher produces area-specific embedding factors, which are then tokenized by treating each time step as a separate token. We replace tokens corresponding to masked areas with a learnable mask token. For each unrecorded area, we also include learnable mask tokens, which allows the model to infer latent factors for unrecorded areas at test time. Inspired by masked autoencoders, which use positional embeddings to preserve the spatial location of image patches [21], we add a brain-area embedding to each token to encode its anatomical location. Although area information is already added in the cross-attention stitcher, adding it to mask tokens is crucial to ensure that the model can distinguish which brain area each token represents. Temporal information is encoded using rotary positional embeddings (RoPE) [45, 2].

**Transformer encoder.** To process the sequence of tokens from multiple areas and animals, we utilize an encoder-only transformer architecture composed of standard transformer blocks followed by a linear layer [52]. By computing interactions between all pairs of tokens in the input sequence, the self-attention mechanism of the transformer encoder allows the model to capture dependencies across brain areas and time steps. The encoder outputs latent factors for all brain areas of interest, including those that were unrecorded in a given session.

**Generalized linear read-out stitcher.** A generalized linear read-out stitcher maps the latent factors back to neural activity using a linear layer followed by an exponential activation. This component is kept intentionally simple to preserve the interpretability of latent factors, ensuring that those for unrecorded areas closely reflect actual neural activity. Its parameters are specific to each session and brain area. The number of latent factors per area is chosen based on the maximum linear dimensionality of that area's smoothed neural activity across sessions (see Appendix A.5).

## 3.2 Masking scheme

To enable the model to infer missing activity from unrecorded areas, we utilize an inter-area masking scheme: in each training batch, we randomly select a masking percentage between 0% and 60%,

and mask out that proportion of the recorded brain areas. We then train the model to reconstruct their activity. Since our goal is to predict activity in unrecorded brain areas, inter-area masking alone is insufficient for generalization. To address this, we introduce additional loss terms designed to improve the model's ability to infer activity in unrecorded areas.

### 3.3 Loss function

The loss function consists of three components: the reconstruction loss, consistency loss, and regularization loss. The **reconstruction loss** is a standard Poisson negative log-likelihood and is used to evaluate the accuracy of predicted firing rates against observed spike counts. The **consistency loss** constrains each brain area's embedding factors to preserve a stable correlation structure between factors across sessions by penalizing deviations from a session-averaged target correlation (see Appendix A.3 and A.4 for details). This constraint helps the model generalize to unrecorded areas in a given session by encouraging the embedding factors to encode information predictive of all observed areas across sessions. Importantly, the consistency loss is applied to embedding factors (see Fig. 1C for definition), which are related to the neural activity through learned nonlinear transformations. That is, we do not assume stable correlation between areas at the level of raw spike data or latent factors. Finally, the **regularization loss** penalizes rapid temporal fluctuations in the latent space, promoting smoother latent dynamics. Formal definitions of the consistency and regularization losses can be found in Appendix A.3.

## 4 Experiments

Our approach builds upon two core assumptions: (1) neural activity in each brain area lies on a *low-dimensional* manifold, allowing a fixed set of low-dimensional latent factors to explain most of the variance in the high-dimensional neural data; and (2) a *consistent and potentially nonlinear mapping* exists between the latent dynamics of different brain areas. To test our proposed model, we first apply it to a synthetic dataset constructed to satisfy both assumptions. We then evaluate its performance on two large-scale Neuropixels datasets spanning multiple animals and brain areas, where these assumptions are expected to hold approximately.

In the synthetic data, ground truth firing rates from unrecorded areas are available. For the real data, *we treat one recorded area per session as unrecorded* by holding it out during training, allowing evaluation during test with ground-truth data. We measure how well the inferred latent factors capture dynamics in the held-out area by training, during the test phase, a generalized linear model (GLM) to predict firing rates for that area from its inferred factors. Note that these GLMs cannot be learned during training for the held-out areas because their spike data are not made available during training.

All trials used for evaluating the latent factors come from a test set that was excluded during NeuroPaint training. We split this test set into 60% training and 40% testing trials for cross-validation of the GLM performance. Performance is quantified using deviance fraction explained (DFE), a normalized goodness-of-fit metric (see Appendix A.7). We use DFE as an intuitive, bounded metric where 1 indicates perfect prediction, 0 matches a null model using the average firing rate, and negative values indicate worse-than-null performance. However, when the null model closely matches the ground truth, DFE can be low even if the model performs well. Conversely, DFE values near 1 may reflect overfitting—not in the conventional sense of poor generalization across data splits, but in that the model may capture high-frequency spike noise rather than meaningful structure in the firing rates.

### 4.1 Datasets

**Synthetic dataset.** To generate synthetic neural data, we simulate an autonomous recurrent neural network (RNN) comprising five brain areas, each with 200 RNN units (see Appendix A.6). Units within each area are densely connected (all-to-all), while inter-area connections are sparse (1%). The recurrent weights induce chaotic dynamics, resulting in trial-to-trial variability driven by different initial conditions. Counterintuitively, the chaotic dynamics generated by such randomly connected networks are often low-dimensional—occupying a subspace whose dimensionality is much smaller than the number of RNN units [11]. For each synthetic session, spike trains are produced using a session-specific GLM with sparse weights (2%), mapping RNN activity to spikes. This setup yields distinct neuron populations per session, each arising from partially overlapping subsets of RNN units.

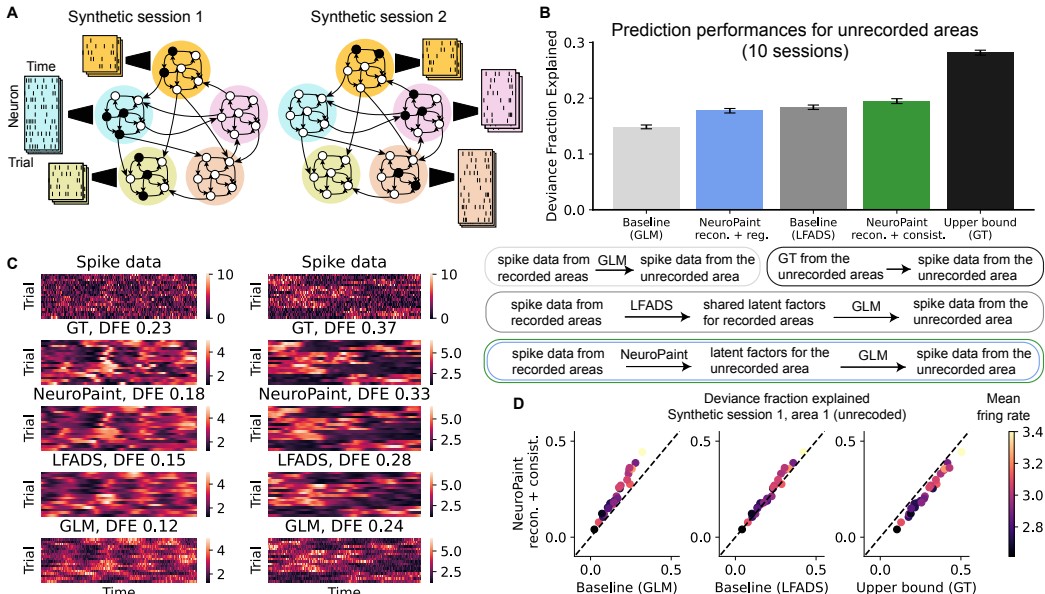

Figure 2: *Quantitative and qualitative comparison of NeuroPaint with baseline models on synthetic datasets.* **(A)** Synthetic spike trains are generated across 10 sessions using a shared underlying RNN, with five brain areas of 200 units each. Spike trains are generated using session- and area-specific GLMs with sparse, random weights, allowing partially overlapping RNN units to contribute differently across sessions. **(B)** Prediction performance for synthetic neurons in unrecorded areas across 10 sessions. The bar plot displays the mean DFE and one standard error across neurons for each method, while the flow diagram shows the computation of DFE. **(C)** Spike data and ground truth firing rates (GT) for two example synthetic neurons, compared with firing rate predictions from NeuroPaint (trained with reconstruction and consistency losses) and the two baselines (GLM and LFADS). **(D)** Prediction performance for synthetic neurons from area 1 (unrecorded) in session 1. Each dot represents a neuron, showing DFE achieved by NeuroPaint (trained with reconstruction and consistency loss), the two baselines, and the upper bound. Dots are color-coded by each neuron's mean firing rate.

In each session, $3 \sim 4$ out of 5 areas are designated as "recorded," and the rest as unrecorded. Each area contains $20 \sim 60$ simulated neurons per session.

**IBL dataset.** We use the International Brain Laboratory (IBL) brain-wide map dataset [28] (Fig. 3A, B). This dataset consists of Neuropixels recordings collected from 12 labs which utilize a standardized experimental pipeline. The recordings target 279 brain areas across 139 adult mice performing the same visual decision-making task. The probe was localized after the experiments using reconstructed histology and the brain areas were annotated. We utilize trial-aligned, spike-sorted data from 20 mice (1 session per mice), and select 8 brain areas for analysis: PO, LP, DG, CA1, VISa, VPM, APN, and MRN (see Appendix A.1 for full area names). From these recordings, we have a total of 21568 neurons for training and evaluation. We bin the spike trains using 10 ms windows and we fix the trial-length to 2 seconds (200 time bins). Trials are categorized into two types: left choice and right choice. Each of the 20 selected sessions has 3 to 7 brain areas simultaneously recorded. We randomly hold out one recorded brain area per session for evaluation.

**MAP dataset.** The Mesoscale Activity Project (MAP) dataset records neural activity underlying memory-guided movement in mice using Neuropixels probes targeting motor cortex, thalamus, midbrain, and hindbrain regions, spanning 293 cortical and subcortical structures [10] (Fig. 3E, F). It includes 173 sessions from 28 mice. We use trial-aligned, spike-sorted data from 40 sessions across 16 mice, binned in 10 ms windows over fixed 4-second trials (400 time bins). Analyses are restricted to two trial types, *hit left* and *hit right*, which are also used to compute the consistency loss. For our analysis, we focus on 8 brain areas, 6 highly relevant to the task [19, 29, 18, 49, 9] and 2 orbital areas: ALM, lOrb, vlOrb, Pallidum, Striatum, VAL-VM, MRN, and SC (see Appendix A.1 for full area names). Each of the selected sessions has 4 to 6 brain areas simultaneously recorded. We randomly hold out one recorded brain area per session for evaluation.

## 4.2 Baselines

We compare NeuroPaint against linear and non-linear baselines. We use a standard generalized linear model (GLM) for our linear baseline. For our non-linear baseline, we compare to Latent Factor Analysis via Dynamical Systems (LFADS) [33, 46], a sequential variational autoencoder that has demonstrated strong performance in capturing latent neural dynamics. We utilize a re-implemented version of LFADS [38] for all our analyses, as discussed below.

**GLM.** We implement GLMs that map from instantaneous spike activity in all the recorded areas to instantaneous spike activity in an unrecorded (in the synthetic dataset) or held-out (in the Neuropixels datasets) area. The GLM consists of a linear layer followed by an exponential nonlinearity, and is trained using the Poisson negative log-likelihood loss. We fit a separate GLM for each session and each held-out area.

**LFADS.** We re-implement multi-session LFADS [46, 33] to model shared neural dynamics across multiple animals. The objective of LFADS is to maximize the likelihood of observed neural activity by reconstructing spike trains from a low-dimensional latent dynamical system. For each session's neural population, multi-session LFADS learns a linear projection layer (read-in stitcher) to embed the neural activity into a shared latent space. It also learns a corresponding set of read-out stitchers to map the latent representations back to neural activity space. Unlike NeuroPaint, LFADS embeds activity from all recorded areas into a shared latent space but cannot infer latent dynamics for unrecorded areas, as it is only trained to reconstruct observed data and does not use masked modeling to predict missing areas. Most importantly, LFADS lacks area-specific latent factors, which limits its interpretability in multi-area regimes. We apply principal component regression to select the latent dimensionality and to pre-condition both the read-in and read-out stitchers [33] (see Appendix A.5). Further details on architecture, weight initialization, and implementation can be found in Appendix A.10 and A.11.

In summary, we evaluate the ability of each approach to predict neural activity in unrecorded areas, by using a supervised GLM (trained during the test phase) to predict neural activity in held-out areas. Specifically, we compare the predictive power of three types of inputs to the GLM: (1) spike activity from recorded areas (corresponding to the GLM baseline), (2) LFADS latent factors shared across recorded areas, and (3) NeuroPaint's area-specific latent factors for the unrecorded area. Although we are comparing NeuroPaint's performance with these other baselines, they are not comparable in the important sense that NeuroPaint is the only approach that explicitly learns separate latent factors for each brain area, including those unrecorded ones, whereas the two baselines cannot infer area-specific latent dynamics.

# 5 Results

## 5.1 Synthetic dataset

We evaluate NeuroPaint on a synthetic dataset with known ground truth firing rates, enabling computation of an upper-bound DFE using spike data in the unrecorded areas and true rates. The model is trained on recorded areas from 10 synthetic sessions and evaluated on its ability to infer single-neuron firing rates in unrecorded areas. The architecture follows Section 3.1, excluding hemisphere embeddings due to the absence of hemispheric structure in the simulation. We compare two model variants, one trained with reconstruction plus regularization loss, the other with reconstruction plus consistency loss; combining all three losses (not shown) results in over-regularization and performance degradation below baseline. This is likely due to the fast, high-firing-rate dynamics in the synthetic data; as suggested by an additional experiment on a separate low-firing-rate synthetic dataset, where the full model (all three losses) outperforms the ablated variant lacking one loss term (see Appendix A.6 and Supplementary Fig. 2). We compare each model against GLM and LFADS baselines and the upper bound. The consistency-loss variant outperforms the two baselines and the regularized model, and is the closest to the upper bound (Fig. 2B, D). As shown in Fig. 2C, NeuroPaint more accurately captures temporal structure than GLM, which tends to overfit to Poisson noise, and slightly outperforms LFADS in recovering fine-grained features of the ground truth firing rates. These results show that under the idealized conditions of the synthetic data, *i.e.* low-dimensional activity and consistent cross-area mapping, NeuroPaint can accurately infer neural dynamics in unrecorded areas.

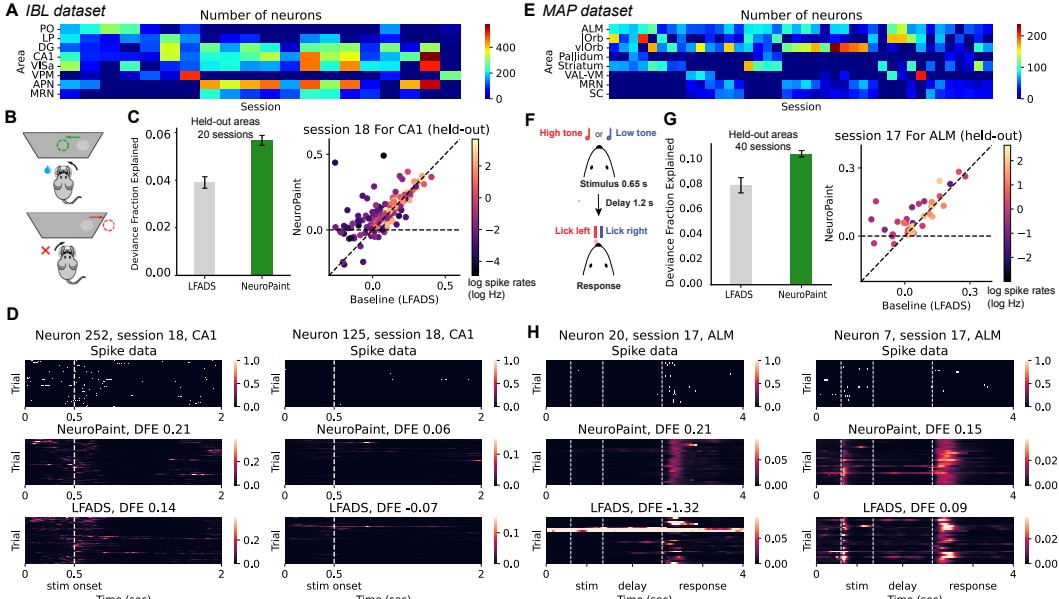

Figure 3: *Quantitative and qualitative comparison of NeuroPaint with baseline models on IBL and MAP datasets.* **(A)** Number of neurons recorded per area in 20 IBL sessions. **(B)** In the IBL experimental setup, mice perform a visual decision-making task by turning a wheel to the left or right to indicate whether the stimulus is presented on the left or right screen [28]. **(C)** Left: The mean and standard error of DFE for LFADS and NeuroPaint, calculated for neurons from held-out areas pooled across 20 IBL sessions, excluding the worst 1% of neurons for visualization; see Fig. S1A for results without exclusion. Right: Per-neuron DFE for the held-out area (CA1) of session 18. **(D)** Spike trains and predicted firing rates for two example neurons, compared between NeuroPaint and LFADS predictions. **(E)** Number of neurons recorded per area in 40 sessions from MAP dataset. **(F)** The MAP dataset records neural activity from mice performing a memory-guided directional licking task, in which they are instructed to lick left or right according to auditory tones presented before a fixed delay period. [9]. **(G, H)** show similar content to **(C, D)**, but with neurons from held-out areas pooled across 40 MAP sessions (see Fig. S1B for results with the worst 1% exclusion).

## 5.2   IBL and MAP datasets

In the Neuropixels datasets, although we have ground truth spiking data, the absence of ground truth firing rates precludes computation of an upper-bound DFE. While absolute DFE values may appear lower than those observed in the synthetic data, they reflect the intrinsic performance ceiling set by the low and sparsely modulated firing rates typical of real neuronal activity. Rather than indicating suboptimal performance, these values underscore the challenge of the task and the robustness of the model's performance under realistic biological constraints.

We train two separate NeuroPaint models, one for the 20 IBL sessions and one for the 40 MAP sessions. In both datasets, we compare NeuroPaint to LFADS, which is the highest performing baseline for single-neuron prediction accuracy in held-out areas. The GLM baseline performs poorly across both datasets due to its inability to model non-linear, cross-area interactions in real neural data [56, 54], and is therefore omitted from the comparison (results are provided in Appendix A.9).

Across held-out areas, NeuroPaint outperforms LFADS by a large margin in both the IBL and MAP datasets (Fig. 3C, G; Fig. S1). Example raster plots of individual neurons further show that NeuroPaint captures structured, single-trial dynamics that LFADS fails to recover (Fig. 3D, H). In particular, LFADS exhibits large outlier errors for some neurons (e.g., neuron 20 in Fig. 3H), whereas NeuroPaint delivers more stable and accurate firing rate predictions. In the neural data, which has relatively low firing rates compared to the synthetic data of Fig. 2, the combination of all three loss terms performs best: ablation experiments show that removing either the regularization or the consistency loss degrades performance relative to the full NeuroPaint model (see Appendix A.13, Supplementary Fig. 1). Moreover, increasing the number of parameters in the LFADS model does not close the performance gap to NeuroPaint (see Appendix A.15), suggesting that NeuroPaint's advantage is not explained by model size alone. These results demonstrate that NeuroPaint not

only infers interpretable, area-specific latent factors for unrecorded brain regions, but also achieves state-of-the-art predictive performance.

### 5.3 Interpretable area-specific latent dynamics revealed by NeuroPaint

We use the MAP dataset to illustrate that NeuroPaint generates consistent and interpretable area-specific latent dynamics across trials and sessions. See similar results for IBL dataset in Appendix A.14.

**Preprocessing.** To highlight dynamic structure over static offsets, we subtract the temporal mean of each trial from each latent factor before computing correlations or visualizing dynamics in Fig. 4. This preprocessing step helps mitigate spurious correlations driven by differences in the latent factor's non-zero temporal means.

**Consistent and context-dependent latent dynamics.** We evaluate (1) the consistency of latent factors across trials for both recorded and unrecorded areas, and (2) the context-dependent variability of latent factors that reflects distinct behavioral conditions, such as *hit left* vs. *hit right* trials where mice respond by licking in different directions. We find that the inferred area-specific latent factors exhibit consistent temporal structure across trials within the same behavioral context (Fig. 4A), even in sessions where the corresponding area is unrecorded (Fig. 4B). Additionally, the latent factors exhibit clear context-dependent variability that captures task-relevant differences; for instance, the latent factors for area SC show distinct patterns between hit left and hit right trials (Fig. 4A). To quantify the consistency and context-dependent variability, we compute the Pearson correlation between latent factors across all trial pairs, averaged over brain areas (see Appendix Table 5). First, latent factors from the same brain areas exhibit positive correlations across trials, regardless of whether those areas were recorded or not in a given session. Second, trial pairs of the same type have higher correlations than those of different types, suggesting that the latent factors capture context-dependent variability. (see Appendix A.16 for more details)

To directly evaluate whether the latent factors capture stimulus- or behavior-dependent variability, we performed decoding analysis from the latent factors. We found that the latent factors in most areas reliably support decoding for both stimulus and behavioral choice, demonstrating that they encode meaningful, task-relevant information. (see Appendix A.17 for more details)

**Inferring area-to-area interactions from inpainted neural data.** We demonstrate that NeuroPaint's inferred latent factors enable novel analyses of area-to-area interactions during behavior. As shown in Fig. 4C, the latent factors from both recorded and unrecorded areas in a single trial reveal rich and distinct temporally structured activity across the 8 selected brain areas. With a fully inpainted neural picture at a single-trial resolution across 40 sessions, NeuroPaint enables analyses on area-to-area interactions that were previously infeasible with Neuropixels recordings, which lack simultaneous coverage of all areas. As one such analysis, we quantify representational similarity across all pairs of brain areas. For each area and each trial period (stimulus, delay, and response), we first compute a time-by-time representational dissimilarity matrix (RDM), defined as one minus the pairwise correlation between flattened latent factors. We then measure area-to-area similarity by correlating the upper-triangular elements of these RDMs, averaged across trials over multiple sessions (Fig. 4D). This analysis reveals that inter-area relationships evolve across behavioral epochs: similarity is relatively low during stimulus and delay periods but increases during the response period, which is consistent with previous findings of increased inter-area coordination during motor output [9]. While Chen et al. [9] were limited to analyzing interactions between ALM and a few simultaneously recorded areas, NeuroPaint overcomes this limitation by enabling comparisons across all area pairs, even including unrecorded areas, via latent inpainting. Furthermore, we observe that the two orbital areas (lOrb and vlOrb) exhibit high mutual similarity, reflecting their shared anatomical and functional roles [32]. In contrast, these areas show weak similarity to the remaining brain areas, even during the response period, which aligns with prior knowledge about the involvement of the selected non-orbital areas in task-relevant computations [19, 29, 18, 49, 9].

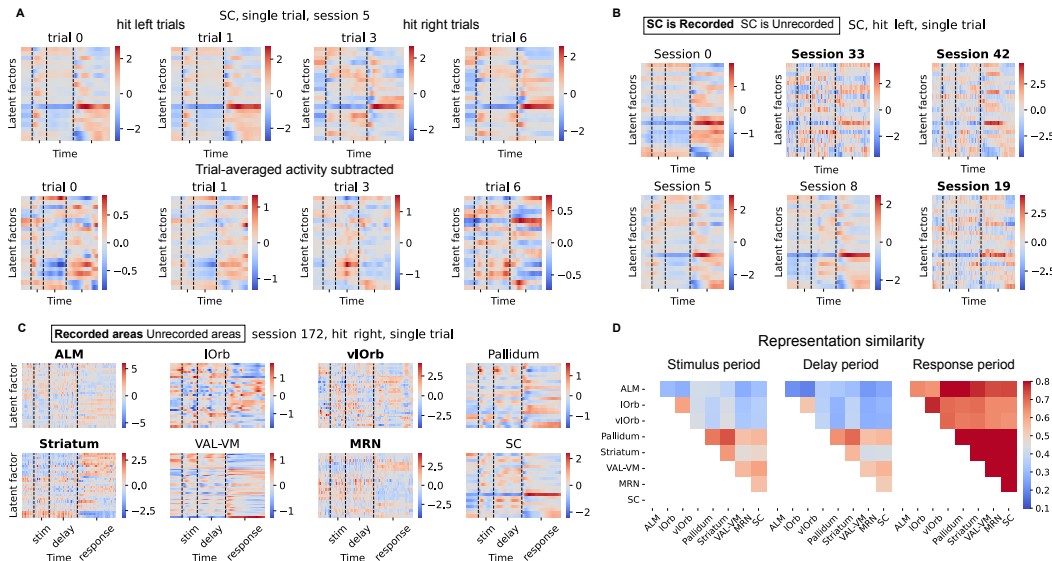

Figure 4: *NeuroPaint's area-specific latent factors are consistent, capture context-dependent variability, and enable inference of area-to-area interactions.* **(A)** Inferred latent factors for the Superior Colliculus (SC), an unrecorded area, during example trials across the *hit left* and *hit right* conditions in a single session. The bottom row shows the same latent factors with the trial-averaged activity subtracted. **(B)** Inferred single-trial latent factors for SC (both recorded and unrecorded) across six sessions. **(C)** Latent factors for all eight areas (both recorded and unrecorded) in a *hit right* trial from a single session in the MAP dataset. **(D)** Representation similarity analysis (RSA) of latent factors across eight brain areas (recorded and unrecorded) for the stimulus, delay, and response periods in the MAP dataset, with values averaged across trials and pooled over sessions.

## 6 Discussion

In this work, we introduce NeuroPaint, a masked transformer-based model that infers the dynamics of unrecorded brain areas by leveraging shared activity structure across animals. Our experiments on synthetic and large-scale Neuropixels datasets show that NeuroPaint outperforms both linear and non-linear baselines, including LFADS and GLMs, in predicting single-neuron activity in held-out areas. Beyond predictive accuracy, NeuroPaint produces interpretable latent dynamics that are consistent across sessions and sensitive to behavioral context, enabling new forms of cross-area and cross-session analysis that were inaccessible with existing tools.

Despite these strengths, several limitations remain. First, training NeuroPaint is computationally expensive, as the self-attention mechanism scales quadratically with the number of brain areas and time steps. Future work could address this by incorporating sparse or low-rank attention mechanisms [8]. Second, the latent dimensionality for each brain area is not manually selected, it is estimated via a principled procedure (see Appendix A.5). Nonetheless, the procedure involved design choices that introduce some arbitrariness, and we find that the selected number of latent factors often exceeds the intrinsic dimensionality of the predicted firing rates in each brain area (see Appendix A.18). More automated hyperpameter selection methods, such as Population Based Training [26], could reduce this arbitrariness. Finally, while we demonstrate proof-of-concept results on two datasets, our experiments involve a limited subset of brain areas and sessions relative to the full datasets. Scaling to hundreds of sessions is straightforward, but extending to hundreds of brain areas will require architectural innovations and improved training strategies. Nonetheless, this represents a critical step toward building truly brain-wide models.

## Acknowledgments and Disclosure of Funding

This project was supported by the Simons foundation (543017), National Institutes of Health (1RF1DA056397, 1K99NS144599), the Gatsby Foundation. This work used NCSA Delta GPU at National Center for Suppercomputing Applications through allocation OTH250001 from the Ad-

vanced Cyberinfrastructure Coordination Ecosystem: Services & Support (ACCESS) program, which is supported by U.S. National Science Foundation grants (2138259, 2138286, 2138307, 2137603, and 2138296). We are grateful to Susu Chen, Thinh Nguyen, Nuo Li for discussions about accessing the MAP dataset. We thank Mehdi Azabou for discussion about the design of cross-attention stitcher. We thank Manuel Beiran and David Clark for discussion about neural dynamics in random recurrent neural networks used in the synthetic dataset.

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

# A Technical Appendices and Supplementary Material

## A.1 Dataset details

This section provides the full anatomical names and corresponding acronyms of the brain areas analyzed throughout the paper.

**IBL dataset.** The posterior thalamus (PO), lateral posterior nucleus (LP), dentate gyrus (DG), hippocampal CA1 (CA1), anterior/anteromedial visual area (VISa), ventral posteromedial nucleus (VPM), anterior pretectal nucleus (APN), and midbrain reticular nucleus (MRN).

**MAP dataset.** The anterior lateral motor cortex (ALM), orbital area, lateral part (lOrb), orbital area, ventrolateral part (vlOrb), pallidum combined with globus pallidus, external segment (Pallidum), striatum combined with caudoputamen (Striatum), ventral anterior-lateral complex of the thalamus combined with ventral medial nucleus of the thalamus (VAL-VM), midbrain reticular nucleus (MRN), and superior colliculus, motor related (SC). ALM is identified following methods described in [47].

## A.2 Generative process

For the $N_r$ neurons in area $r$, we construct neural data tokens $Z_{X_r} \in \mathbb{R}^{N_r \times T^{*'}}$ by concatenating neural acivity $X_r \in \mathbb{R}^{N_r \times T}$ with area, hemisphere, and unit embeddings: $Z_{\text{area}} \in \mathbb{R}^{D_a}, Z_{\text{hemi}} \in \mathbb{R}^{D_h}, Z_{\text{unit}} \in \mathbb{R}^{D_u}$, where $T^{*'} = T + D_a + D_h + D_u$. We then compute keys $(K_r)$ and values $(V_r)$ for the cross-attention module using linear projections $W_K, W_V \in \mathbb{R}^{T^* \times T^{*'}}$. A shared set of learnable latent tokens $Z_0 \in \mathbb{R}^{P \times T^*}$ acts as the query $Q$. The resulting cross-attention output is projected via a multi-layer perceptron to obtain embedding factors $Z_{\text{embed},r} \in \mathbb{R}^{P \times T}$.

For each trial, a masking proportion $p$ is chosen from a uniform distribution from 0 to 0.6. If $p <= 0.05$, then no areas are masked, otherwise, if the trial has recordings from $R$ areas (where held-out areas are considered unrecorded), then Ceiling$[p * R]$ areas are randomly chosen to be masked. A variable $M_r$ is set to 1 for masked areas and 0 for unmasked areas. The embedding factors $Z_{\text{embed},r}$ are tokenized by treating each timestep as a token, resulting in $T$ tokens $Z_r^{\text{token}} \in \mathbb{R}^{T \times H}$, where an MLP projects each token from $\mathbb{R}^P$ to $\mathbb{R}^H$. For masked or unrecorded areas, these neural data tokens are replaced by a learned mask token $Z_{\text{mask}}$. We define a binary indicator $U_r \in \{0, 1\}$ to denote whether area $r$ is unrecorded. Each token is then concatenated with a new learnable area embedding $Z'_{\text{area}} \in \mathbb{R}^{D_{a'}}$, producing the input tokens $Z_r^{\text{input}} \in \mathbb{R}^{T \times H'}$, $H' = H + D_{a'}$. These token sequences are passed through a transformer (with RoPE applied to queries and keys), and the outputs (also $\in \mathbb{R}^{T \times H'}$) are projected via $W_r^{\text{latent}} \in \mathbb{R}^{H' \times D_r}$ to obtain latent factors $Z_{\text{latent},r} \in \mathbb{R}^{T \times D_r}$. Finally, firing rate predictions $\hat{X}_r$ are computed via a linear readout $W_r^{\text{rate}} \in \mathbb{R}^{D_r \times N_r}$ followed by exponentiation.

The training procedure is as follows:

| **Cross-attention stitcher** | | **Remaining modules of NeuroPaint** | |
|---|---|---|---|
| $Z_{X_r}$ | $= [X_r, Z_{\text{area}}, Z_{\text{hemi}}, Z_{\text{unit}}]$ | $Z_r^{\text{token}}$ | $= \text{Tokenizer}(Z_{\text{embed},r})$ |
| $K_r, V_r$ | $= W_K(Z_{X_r}^T), W_V(Z_{X_r}^T)$ | $\delta_r$ | $= \begin{cases} 1 & \text{if } M_r = 0 \wedge U_r = 0 \\ 0 & \text{otherwise} \end{cases}$ |
| $Q$ | $= Z_0$ | $Z_r^{\text{input}}(t)$ | $= [(1 - \delta_r)Z_{\text{mask}} + \delta_r Z_r^{\text{token}}(t), Z'_{\text{area}}]$ |
| $Z_{\text{cross-attn},r}$ | $= \text{Cross-Attention}(Q, K_r, V_r)$ | $Z_{\text{latent},r}$ | $= (\text{Transformer}(\text{RoPE}(Z_r^{\text{input}})))W_r^{\text{latent}}$ |
| $Z_{\text{embed},r}$ | $= \text{MLP}(Z_{\text{cross-attn},r})$ | $\hat{X}_r^T$ | $= \text{Poisson}(\exp(Z_{\text{latent},r}W_r^{\text{rate}}))$ |

$$(1)$$

The model assumes a Poisson emission process with time-varying rates. For notational simplicity, we describe the generative model using data from one trial in one session. Symbols with the subscript $r$ denote area-specific parameters, while those without $r$ are shared across areas. Most parameters are shared across sessions, except for $Z_{\text{unit}}$ and $W_r^{\text{rate}}$.

## A.3 Loss function details

The consistency and regularization losses are formally defined below.

To ensure the embedding factors maintain a stable correlation structure across sessions and penalize deviations from a session-averaged target, we introduce the following consistency loss.

**Definition A.1 (Consistency loss).** *Let $Z_{embed,r}^{(b)}(t) \in \mathbb{R}^P$ denote the embedding factors for area $r \in [R]$ at time $t$ under trial type $b \in [B]$. Define $K_{model}^{(b)}(r, r') \in \mathbb{R}^{P \times P}$ as the Pearson correlation matrix between $Z_{embed,r}^{(b)}(t)$ and $Z_{embed,r'}^{(b)}(t)$ computed across time after aggregating time steps over trials of type $b$ within a batch. The corresponding target correlation matrix $K_{target}^{(b)}(r, r') \in \mathbb{R}^{P \times P}$ is obtained by averaging the correlation between $Z_{embed,r}^{(b)}(t)$ and $Z_{embed,r'}^{(b)}(t)$ across time steps in trials of type $b$, aggregated over sessions. The consistency loss is then defined as follows.*

$$\mathcal{L}_{\text{consist.}} = \sum_{b}^{B} \sum_{r'=1}^{R} \sum_{r=1}^{R} 1 - \cos\left(\text{vec}\left(K_{\text{target}}^{(b)}(r, r')\right), \text{vec}\left(K_{\text{model}}^{(b)}(r, r')\right)\right), \tag{2}$$

*where $\cos(\cdot, \cdot)$ denotes the cosine similarity, and $vec(\cdot)$ is a vectorization operator that flattens the correlation matrix, using only the upper triangular elements when $r = r'$.*

The consistency loss encourages each area's embedding factors to encode information predictive of all areas observed across sessions, in addition to those recorded in the current session, thereby improving generalization to unrecorded areas. In practice, we approximate the session-averaged target $K_{\text{target}}^{(b)}(r, r')$ using a buffer of past batch correlations, computed with an exponential moving average (EMA) of the cross-attention stitcher to improve stability (Appendix A.4) [48].

Under the assumption that neural dynamics evolve smoothly over time, we define the following regularization loss.

**Definition A.2 (Regularization loss).** *Let $Z_{latent,r}(t) \in \mathbb{R}^{D_r}$ denote the latent factors at time $t$ for area $r \in [R]$ and let $Z_{latent,r}(t, i)$ denote its $i$-th component. The regularization loss is defined as follows.*

$$\mathcal{L}_{\text{reg.}} = \sum_{t=1}^{T-1} \sum_{r=1}^{R} \sum_{i=1}^{D_r} |Z_{\text{latent},r}(t+1, i) - Z_{\text{latent},r}(t, i)|. \tag{3}$$

The regularization loss penalizes abrupt changes across consecutive time steps in the latent space, thereby improving temporal smoothness of the learned latent factors.

The relative weighting of the loss terms was selected to ensure that the reconstruction loss remains the primary driver during training, while the consistency and regularization terms serve as auxiliary constraints. Specifically:

- The reconstruction loss is scaled by a weight of 1 and normalized by the number of timesteps and neurons.
- The consistency loss is scaled by a weight of 1 and normalized by the number of area pairs.
- The regularization loss is scaled by a weight of 0.1 and normalized by the number of timesteps and latent factors.

### A.4 Exponential moving average (EMA) of the cross-attention stitcher

To stabilize the correlation target calculated from the area-specific embedding factors, we maintain an EMA version of the cross-attention stitcher during training. At each iteration, the parameters in the EMA version are updated as a weighted sum of their previous values and the current weight of the cross-attention stitcher in the main model, controlled by a decay rate $\alpha$. Specifically, the EMA parameter $\theta$ is updated as follows:

$$\theta^{EMA} \leftarrow \alpha \theta^{EMA} + (1 - \alpha)\theta \tag{4}$$

To reduce bias in early training when parameter estimates are still unstable, we adapt the decay rate $\alpha$ based on the number of iteration steps $n_{\text{step}}$:

$$\alpha = \min(1 - \frac{1}{n_{\text{step}} + 1}, \alpha_{\text{max}}) \tag{5}$$

which increase $\alpha$ as training progresses. Here $\alpha_{\max} = 0.999$. This ensures faster adaptation in the early iterations and more stable averaging later on. We use the EMA cross-attention stitcher to compute the area-specific embedding factors and then compute the correlation between embedding factors for each batch, and store them in a running buffer. These stored correlation matrices are then averaged to generate the target correlation matrix for each area pair and trial type for the current iteration.

## A.5 Determining the area-specific latent factor dimension in NeuroPaint and LFADS

**NeuroPaint.** To determine the number of latent factors for each brain area, we estimated the dimensionality of smoothed neural activity across sessions. For each area and session, we first smoothed the spike trains with a Gaussian window (standard deviation = 50 ms), concatenated all trials, and subtracted the mean to center the data. We then computed the participation ratio of the resulting activity matrix, which provides a measure of its linear dimensionality [13]. This yielded one dimensionality estimate per area per session. To ensure sufficient capacity, we set the number of latent factors for each brain area to the maximum participation ratio observed across sessions for that area, plus a margin of 10 dimensions. We observed that for most brain areas, the estimated dimensionality saturated as the number of recorded neurons increased, suggesting that the low-dimensional assumption we mentioned at the end of the introduction is approximately satisfied by the neural data. This procedure ensures that NeuroPaint is equipped with enough latent factors to model the full range of dynamics expressed in each area.

| Area | Dim |
|---|---|
| PO | 61 |
| LP | 84 |
| DG | 72 |
| CA1 | 39 |
| VISa | 35 |
| VPM | 43 |
| APN | 58 |
| MRN | 24 |
| ALM | 29 |
| lOrb | 31 |
| vlOrb | 37 |
| Pallidum | 23 |
| Striatum | 31 |
| VAL-VM | 49 |
| MRN | 22 |
| SC | 21 |

Table 1: NeuroPaint latent factor dimensions picked for areas from IBL and MAP datasets.

**LFADS.** The multi-session LFADS [33] uses per-session linear read-in and read-out layers to align neural activity from different sessions into a shared latent space. These session-specific "stitching" layers map observed spike counts to input factors (read-in) and latent factors to firing rates (read-out). The shape of these matrices can vary to match the number of neurons recorded in each session. A shared encoder, generator, and factor matrix are shared across sessions and learned jointly from all sessions. The per-session read-in and read-out matrices are learned using data from only the corresponding session. To promote alignment across sessions, we follow the standard procedure to initialize the weights of the read-in and read-out matrices using principal component regression (PCR) [38]. PCR maps the trial-averaged firing rates from each individual session to the shared principal components across all sessions. This provides the model with an input that is already in a shared subspace and is critical for ensuring a shared set of dynamics is learned. We first reshape the trial-averaged firing rates from all sessions into a matrix of size $(n_{\text{timepoints}} \times n_{\text{conditions}}) \times (n_{\text{sessions}} \times n_{\text{neurons}})$, and PCA is applied to identify a set of global principal components (PCs). We find the number of PCs needed to explain 90% of the variance, which defines the dimensionality of the latent factors used in LFADS. Each session's data is then projected onto this global PC space via Ridge regression, and the

resulting weights and biases from the trained regression model are saved to be used to initialize the read-in weights and biases in LFADS.

## A.6 Details of the simulated recurrent neural network in the synthetic dataset

To generate synthetic spike data with multi-area dynamics, we simulated a continuous-time recurrent neural network (RNN). This RNN is composed of 5 areas, each area contains 200 units. Each unit has a $\tanh$ nonlinearity and with a time constant $\tau = 25$ ms. We simulate the network using a timestep of $\Delta t = 10$ ms. Unit activity evolves according to the following update rule:

$$h_{t+1} = (1 - \beta)h_t + \beta \tanh(W h_t) \tag{6}$$

Here $\beta = \frac{\Delta t}{\tau}$. $W$ is the recurrent connectivity between units. $h_t$ denotes unit activities at time step $t$.

The network receives no external input, its activity is driven entirely by autonomous recurrent dynamics. Each recurrent connection has a weight sampled from a normal distribution $W_{ij} \sim \mathcal{N}(0, g^2/N)$, here $W_{ij}$ is the weight between unit $i$ and unit $j$, $N = 1000$ is the total number of units in the network, we pick $g = 3$ so that we will have chaotic circuit dynamics [40]. Connectivity between areas is sparse: each pair of units are connected with $1\%$ probability. Units within the same area have all-to-all connectivity.

For each trial, we randomly initialize the unit activity in the network. Due to chaotic dynamics, each trial will have qualitatively different dynamics. We simulate trials from the same RNN across synthetic sessions.

For each synthetic session, we simulate neuronal spike trains from the unit activity using GLMs. Each session has its own set of synthetic neurons, with the number of neurons in each area sampled uniformly from the range $[20, 60]$. The number of trials in each synthetic sessions are sampled uniformly from the range $[200, 300]$.

Each area-specific and session-specific GLM first projects the unit activity onto neuronal instantaneous log firing rates using a sparse linear layer ($2\%$ sparsity). The log firing rates for each synthetic neuron are then scaled to a fixed range ([0,2]). Next, we pass the log rates to an exponential nonlinearity and a Poisson emission process. For each session, we held-out spike activities generated from 1-2 RNN areas as unrecorded.

In summary, partially overlapped RNN units contribute to the simulated spike trains across synthetic sessions.

## A.7 Deviance fraction explained (DFE)

Deviance fraction explained is a metric that ranges from 0 to 1, where a larger value indicates a better model fit. It is defined as:

$$1 - \frac{D_{\text{model}}}{D_{\text{null}}}$$

where the deviance terms are given by:

$$D_{\text{model}} = \log p(\mathbf{x}_{1:T}|M_{\text{sat}}) - \log p(\mathbf{x}_{1:T}|M)$$
$$D_{\text{null}} = \log p(\mathbf{x}_{1:T}|M_{\text{sat}}) - \log p(\mathbf{x}_{1:T}|M_{\text{null}})$$

Here, $M_{\text{sat}}$ represents the **saturated model**, while $M_{\text{null}}$ represents the **null model**. Both the saturated and null models assume that the instantaneous spike data is generated from an i.i.d. Poisson distribution. The **saturated model** ($M_{\text{sat}}$) assumes that the mean of the Poisson distribution is given by the instantaneous spike count, while the **null model** ($M_{\text{null}}$) assumes that the mean is given by the time-averaged and trial-averaged spike count.

## A.8 Evaluation on Synthetic Data with Lower Firing Rates (Matching Real Data Statistics)

To better align our synthetic data with the firing rate statistics observed in the real neural recordings, we generated an additional synthetic dataset, where the log firing rates of each neuron were scaled to a fixed range of $[-3, 3]$, in contrast to the range $[0, 2]$ used in the previous synthetic dataset shown in Fig. 2. This adjustment results in lower and more realistic firing rates.

We evaluated the performance of several models for predicting neural activity in the unrecorded brain areas on this synthetic dataset: two baselines (GLM and LFADS) and multiple variants of NeuroPaint (including the linear version described in A.12 and other variants trained with different combinations of loss terms). Overall, we found that NeuroPaint trained with all three loss terms (reconstruction, consistency and regularization) achieved the best performance for this low-rate synthetic dataset (see Fig. S2), unlike for the higher-rate synthetic dataset discussed in the main text (Fig. 2). Empirically, we found that including the consistency loss was particularly important for improving NeuroPaint's performance, enabling it to outperform the baselines.

It is important to note that for the synthetic dataset, the GLM baseline also performs very well, suggesting that a linear model is sufficient to predict neural activity in one area based on data from other areas, i.e. to capture inter-areal communication in this simplified setting. In future work, we plan to construct more complex and biologically realistic synthetic datasets that include strong nonlinear inter-areal dependencies, to better evaluate the advantage of nonlinear models.

### A.9 GLM baseline performance on the IBL and MAP datasets

We evaluated GLM baselines on both the IBL and MAP datasets. For the IBL dataset, the mean deviance fraction explained (DFE) across neurons in the held-out areas was extremely negative ($-3.8 \times 10^{32}$), indicating severe overfitting or model mismatch. Notably, 90.7% of neurons had negative DFE values. For the MAP dataset, the mean DFE was $-5.1 \times 10^{23}$. 63.3% of neurons exhibited negative DFE values.

### A.10 Model and hyperparameter details

**NeuroPaint.** To avoid overfitting, we add dropout (40%) to all the attention layers and a dropout (20%) at the very beginning. Other hyperparameters of the NeuroPaint model is listed in Table 2. Notations for the hyperparameters are introduced in A.2. For synthetic dataset, we use 24 latent factors for each area. For the IBL and MAP dataset, we set the dimensionality of latent factors according to A.5.

| Hyperparameter | Value |
|---|---|
| $D_a$ | 20 |
| $D_h$ | 3 |
| $D_u$ | 50 |
| $T$ | 400 (MAP), 200 (IBL) |
| $T^*$ | 512 |
| $P$ | 48 |
| $H$ | 236 |
| $D_a'$ | 20 |
| $H'$ | 256 |
| number of transformer layers | 5 |

Table 2: Hyperparameters used for NeuroPaint.

**LFADS.** We use a version of LFADS (*lfads-torch*) re-implemented in PyTorch by [38]. The hyperparameters of our multi-session LFADS are provided in Table 3. The shared encoder, generator, and factor matrix comprise about 0.3 million parameters. When including session-specific read-in and read-out matrices, the total number of model parameters increases to 0.4 million, 0.5 million, and 0.6 million for the synthetic, IBL and MAP datasets. The latent factor dimension of multi-session LFADS is chosen via PCR (Appendix A.5). We set the latent factor dimension to 28, 11, and 27 for the synthetic, IBL and MAP datasets. The LFADS model computes KL divergence between posteriors and priors for both initial condition and inferred input distributions, which are added to the reconstruction cost in the variational ELBO. The priors are multivariate normal for the initial conditions and autoregressive multivariate normal for the inferred inputs.

**NeuroPaint-Linear.** The linear, session-specific, and area-specific stitchers take as input the flattened population activity from 3 neighboring bins centered around the target time point. Prior to this, the binned spike counts are smoothed using a 1D Gaussian kernel with a standard deviation of 5

| Hyperparameter | Value |
|---|---|
| Initial Condition Encoder Dimension | 100 |
| Controller Input Encoder Dimension | 100 |
| Controller Input Lag | 1 |
| Controller Dimension | 100 |
| Controller Output Dimension | 6 |
| Initial Condition Dimension | 100 |
| Generator Dimension | 100 |
| Controller Output Prior Temporal Decay Constant | 10 |
| Controller Output Prior Innovation Variance | 0.1 |
| Initial Condition Prior Mean | 0 |
| Initial Condition Prior Variance | 0.1 |
| Dropout Rate | 0.02 |
| Coordinated Dropout Rate | 0.3 |
| Weight Decay | 0.0 |
| Learning Rate | 0.0003 |
| Batch Size | 1024 |

Table 3: Hyperparameters used for multi-session LFADS.

time bins. The stitchers output area-specific embedding factors, which, along with the latent factors, are set to have the same dimensionality as in the original NeuroPaint model. The session-shared, reduced-rank linear transformation applied to the concatenated embeddings has a fixed rank of 20. The population activity in the masked areas, which the model is trained to predict, is also smoothed using a Gaussian kernel with the same standard deviation of 5 time bins.

## A.11 Training details

**NeuroPaint.** We trained our model on 2-12 Nvidia A40 or A100 GPU using AdamW optimizer for 1000 epochs with a learning rate of $(1e^{-3}/256 \times$ global batch size) using a OneCycleLR scheduler. We put a weight decay $0.01$ to avoid overfitting. We utilized a batch size of 16 on each GPU during the training. Global batch size is $16\times$ the number of GPU nodes. We split our dataset based on the session to training, validation, and test set with a proportion of 60%, 20%, and 20%. We saved the model checkpoint based on the validation loss. The 10-session model for the synthetic dataset is trained with 2 GPU nodes in around 6 hours. The 20-session model for the IBL dataset is trained with 4 GPU nodes in around 35 hours. The 40-session model for the MAP dataset is trained with 12 GPU nodes in around 20 hours.

**LFADS.** We train the multi-session LFADS using the AdamW optimizer without a learning rate scheduler, which is the default setting in the *lfads-torch* package. All multi-session LFADS models are trained on a single Nvidia A40 or A100 GPU for 1000 epochs. The best model checkpoint is selected based on the reconstruction performance on the validation set. The 10-session model for the synthetic dataset is trained in under 4 hours. The 20-session LFADS for the IBL dataset takes approximately 10 hours, while the 40-session LFADS for the MAP dataset is trained in about 16 hours.

**NeuroPaint-Linear.** Training details are provided in Appendix A.12. All models are trained on a single NVIDIA RTX 2080 Ti GPU and complete within one hour.

## A.12 NeuroPaint-Linear

We introduce NeuroPaint-Linear, a *linear* variant of the original NeuroPaint model. Designed for simplicity and interpretability.

**Architecture** NeuroPaint-Linear retains the core architectural components of NeuroPaint, comprising three main components:

1. **Linear, session-specific, and area-specific** stitchers, which map population activity to a set of embedding factors.

2. A **low-rank, session-shared** weight matrix, which transforms the concatenated embedding factors (across all areas) into area-specific latent factors, capturing a mapping between the latent dynamics of different brain areas that remains consistent across sessions and animals.

3. **Linear, session-specific, and area-specific** readout layers, which map latent factors to predicted population activity (for all the recorded and unrecorded areas).

While the overall structure closely mirrors that of NeuroPaint, NeuroPaint-Linear introduces three simplifications:

1. The cross-attention readin-stitcher is replaced by a simple linear mapping.

2. The transformer encoder is replaced with a reduced-rank linear transformation, restricting inter-area mappings to linear operations.

3. In contrast to the nonlinear NeuroPaint, where embeddings for masked or unrecorded areas are replaced with a learned mask token after tokenization, we set these embedding factors to zero directly, bypassing the tokenizer.

In addition, NeuroPaint-Linear operates in a local and time-independent fashion: latent factors and the corresponding predicted activity are computed independently at each time step, using only the recorded activity from a short temporal window (3 time bins, *i.e.* $\pm 1$ bin) around that time—without leveraging long temporal context.

**Training details**    Training follows a similar inter-area masking strategy used in NeuroPaint. For each epoch, one recorded area per session is randomly selected and masked from the input. The model is then trained to predict the activity of the masked area.

Optimization is performed using the L-BFGS algorithm in full-batch mode, minimizing the mean squared error computed across all time steps and all neurons in the masked areas over sessions. For each training epoch, L-BFGS is reinitialized from scratch and run until convergence. We train models for each of the MAP and IBL datasets for 50 epochs each (we only show results for the MAP dataset), using the default L-BFGS hyperparameters provided by PyTorch. We chose L-BFGS and mean squared error due to their stability during training and their effectiveness.

**Comparison with standard reduced-rank regression methods**    NeuroPaint-Linear differs fundamentally from standard reduced-rank regression approaches used in encoding models [4, 36] and inter-areal correlational analyses (e.g., communication subspace) [39], both in its objectives and methodology. The key innovations lie in the introduction of a session-shared mapping from the full set of area-specific embedding factors to area-specific latent factors, and the use of an inter-area masking strategy during training. Together, these components enable NeuroPaint-Linear to predict the dynamics of unrecorded brain areas based solely on the activity of recorded areas within the same session.

### A.13  Impact of loss terms on NeuroPaint performance (MAP dataset)

To evaluate the contribution of each designed loss term on real data, we compared the performance of different NeuroPaint variants on the MAP dataset. In line with the synthetic data results in Fig. 2 and Fig. S2, the inclusion of the consistency loss was essential for enabling NeuroPaint to outperform the LFADS baseline. In addition, NeuroPaint trained with all three loss terms performs the best. Surprisingly, NeuroPaint-Linear, a purely linear variant of the model, also performed competitively, comparably with the LFADS baseline, highlighting the strength of the overall NeuroPaint framework and its potential to reveal interpretable inter-areal communication among all areas of interest, including both recorded and unrecorded ones.

Interestingly, we found that the variant of NeuroPaint trained with reconstruction and regularization loss (but without consistency loss) exhibited unstable performance across datasets. For example, while it performed reasonably well on the original synthetic dataset (Fig. 2), its performance degraded substantially on the lower firing rate synthetic data (Fig. S2) and on the MAP dataset (Fig. S3. This suggests that, in the absence of a consistency constraint, regularization can sometimes hinder model

performance—possibly by over-constraining the latent space or penalizing useful variability. These observations underscore the stabilizing role of the consistency loss in guiding the latent representation, especially under more realistic data distributions.

## A.14  Interpretable area-specific latent dynamics in the IBL dataset revealed by NeuroPaint

Similar to results shown in Fig. 4, here we examine the inferred latent factors for the IBL dataset. Unlike the latent factors for SC in the MAP dataset, the latent factors for MRN in the IBL dataset did not show strong trial-type modulations (Fig. S4A). However, in line with results for the MAP dataset, the inferred latent factors in the IBL dataset remain consistent across trials over sessions, regardless of whether a given area was recorded or unrecorded (Fig. S4A, B). We evaluated area-to-area representation similarity based on the inferred latent factors, across three trial periods, defined as three consecutive 0.5-second intervals after stimulus onset (Fig. S4C,D). Consistent with anatomical expectations, DG and CA1, both belonging to the hippocampus, exhibited high similarity with each other but not with other areas. In addition, two thalamic nuclei, PO and VPM, also exhibit high similarity across the three periods. Notably, the overall inter-areal similarity structure changed over time: although coherent sub-networks of areas remained present throughout the trial, they became more fragmented in the later periods, exhibiting both reduced similarity and a narrower set of similar areas. This dynamic change likely reflects a shift in task phase—from visual processing to decision-making and motor execution.

## A.15  LFADS with larger architectures does not outperform NeuroPaint

To address the concerns regarding potential model capacity limitations in LFADS, we trained LFADS models on the MAP dataset with larger architectures (256 units in encoder, controller and generator). We tested the number of latent factors of 243 (matching the number of pooled NeuroPaint latent factors across areas) and 49 (matching the largest single-area NeuroPaint latent factors). To maintain training stability and prevent loss divergence, we reduced the learning rate from $3e^{-4}$ to $2e^{-4}$ here. Training converged successfully under this setting. The DFE for neurons in held-out areas, pooled over 40 sessions, is reported in Table 4. We found that NeuroPaint continued to outperform LFADS with larger architectures in terms of reconstruction accuracy. Interestingly, the higher-capacity LFADS models performed worse than the smaller model used in our main experiments, potentially due to overfitting given the increased number of parameters. These results suggest that the performance gap is not solely attributable to architectural capacity, and that the paradigm introduced in NeuroPaint is the key to its improved performance.

| | LFADS variants | | | NeuroPaint |
|---|---|---|---|---|
| | 27 factors, 100 units (used in Fig. 3G,H) | 49 factors, 256 units | 243 factors, 256 units | (used in Fig. 3G,H) |
| DFE | $0.08 \pm 0.006$ | $-2.93 \times 10^{20}$ | $-9.12 \times 10^{20}$ | $\mathbf{0.10 \pm 0.003}$ |
| DFE (Worst 5% excluded) | $0.10 \pm 0.002$ | $0.08 \pm 0.002$ | $-9.31 \pm 1.31$ | $\mathbf{0.12 \pm 0.002}$ |

Table 4: Performance comparison across LFADS models of different capacities and NeuroPaint, evaluated by DFE (mean±s.t.e. over neurons in the held-out areas).

## A.16  Calculating correlations between trial pairs of latent factors (MAP dataset)

To assess the consistency and trial-type-dependent modulation of inferred latent factors, we computed Pearson correlations between trial pairs of area-specific latent factors. All analyses were performed separately for each brain area.

**Preprocessing.**  For each trial, the latent factor matrix was first mean-subtracted across time for each factor to remove the static offsets. The resulting matrix was then flattened into a vector by concatenating all factors.

**Consistency results in the left block of Table 5.**  For each area, we computed Pearson correlations between flattened latent-factor vectors for every pair of trials, and grouped trial pairs into three categories: 1) Recorded - Recorded: both trials from sessions where the area is recorded 2) Unrecorded -

Table 5: Mean Pearson correlation between latent factors computed from trial pairs pooled across all sessions and averaged over brain areas (mean $\pm$ 1 standard deviation). See Appendix A.16 for calculation details.

| From all sessions | | Trial-average subtracted, within a session | |
|---|---|---|---|
| Trial pairs | Pearson's correlation | Trial pairs | Pearson's correlation |
| Recorded - Recorded | $0.25 \pm 0.05$ | Hit Left | $0.59 \pm 0.28$ |
| Unrecorded - Unrecorded | $0.57 \pm 0.10$ | Hit Right | $0.59 \pm 0.28$ |
| Recorded - Unrecorded | $0.35 \pm 0.06$ | Across Trial Type | $0.53 \pm 0.31$ |

Unrecorded: both trials from sessions where the area is unrecorded 3) Recorded & Unrecorded: one trial from a session where the area is recorded, one from an unrecorded session.

For each area, we computed the averaged correlation within each group. The reported mean was obtained by averaging these per-area values across all areas, and the standard deviation reflects variability across areas.

**Trial-type-dependent modulation results in the right block of Table 5.** To control for the session-to-session variability, we perform the following analysis on trial pairs within each session, treating each session separately, and again also separately for each area.

For each session, we first computed the trial-averaged flattened latent factor by averaging over all *hit* trials. We then subtracted this average from each trial's flattened latent factor.

We calculated Pearson correlations between all pairs of trials within a session and grouped them into three categories, based on the trial types of the trial pair 1) Hit Left 2) Hit Right 3) Across Trial Type: one hit left and one hit right trial.

For each area and session, we averaged correlations within each group. The reported means and standard deviations were computed by averaging over sessions and areas.

## A.17 Decoding choice and stimulus from NeuroPaint latent factors for held-out areas

To evaluate whether the latent factors capture the stimulus and behavioral variability, we performed decoding analysis on the MAP dataset. Specifically, we used logistic regression with cross-validation to decode either stimulus (high/low auditory tone) or choice (lick left/lick right/no lick) from the latent factors for each held-out area, and we compared the balanced accuracy to that decoding from held-out spike data (see Table 6). Stimulus was decoded using summed activity over the last 100 ms of the stimulus period; choice was decoded using summed activity over the last 100 ms of the delay period. Note that due to randomness, no data from VM-VAL was held out in the MAP dataset. As a result, we are unable to report results for this area.

We found that decoding accuracy for inferred latent factors is typically slightly higher than that from held-out spike data, suggesting that the inferred latent factors successfully capture the stimulus/behavioral variability in the brain area. This decoding improvement also suggests that the inferred latent factors provide a denoised and more complete representation than the held-out population spike data subsampled from the area of interest.

| | | ALM | lOrb | vlOrb | Pallidum | Striatum | MRN | SC |
|---|---|---|---|---|---|---|---|---|
| Stimulus (chance level= 0.5) | spike data | 0.60 $\pm 0.12$ | 0.58 $\pm 0.08$ | 0.53 $\pm 0.08$ | 0.65 $\pm 0.07$ | 0.53 $\pm 0.04$ | 0.62 $\pm 0.08$ | 0.66 $\pm 0.08$ |
| | latent factors | 0.71 $\pm 0.08$ | 0.50 $\pm 0.13$ | 0.61 $\pm 0.13$ | 0.80 $\pm 0.13$ | 0.73 $\pm 0.06$ | 0.68 $\pm 0.12$ | 0.76 $\pm 0.09$ |
| Choice (chance level= 0.33) | spike data | 0.40 $\pm 0.15$ | 0.41 $\pm 0.13$ | 0.37 $\pm 0.12$ | 0.46 $\pm 0.17$ | 0.58 $\pm 0.13$ | 0.45 $\pm 0.09$ | 0.52 $\pm 0.21$ |
| | latent factors | 0.50 $\pm 0.14$ | 0.51 $\pm 0.13$ | 0.57 $\pm 0.14$ | 0.68 $\pm 0.12$ | 0.63 $\pm 0.11$ | 0.63 $\pm 0.16$ | 0.70 $\pm 0.16$ |

Table 6: Decoding accuracy from spike data or latent factors for the held-out areas with logistic regression, evaluated by balanced accuracy (mean$\pm$s.t.d. over sessions).

### A.18 Intrinsic dimensionality of predicted firing rates for each brain area from NeuroPaint latent factors

In Table 7, we report the intrinsic dimension of the reconstructed firing rates based on the NeuroPaint latent factors. Specifically, for each recorded area in the MAP dataset, we calculated the participation ratio of the predicted firing rates, which quantifies the effective number of linear dimensions contributing to the variance. Notably, the number of latent factors we used is much larger than the intrinsic dimensionality of predicted firing rates, suggesting that similar performance may be achievable with fewer latent factors. We will explore this in future work.

|  | ALM | lOrb | vlOrb | Pallidum | Striatum | VAL-VM | MRN | SC |
|---|---|---|---|---|---|---|---|---|
| Number of latent factors | 29 | 31 | 37 | 23 | 31 | 49 | 22 | 21 |
| Intrinsic dimensionality (mean±s.t.d.) | 4.71 ±1.45 | 4.82 ±1.40 | 4.21 ±1.43 | 3.51 ±1.36 | 4.19 ±1.53 | 5.63 ±1.40 | 3.90 ±0.98 | 4.50 ±1.05 |

Table 7: Intrinsic dimensionaliy of predicted firing rates reported against the number of latent factors from NeuroPaint for the MAP dataset.

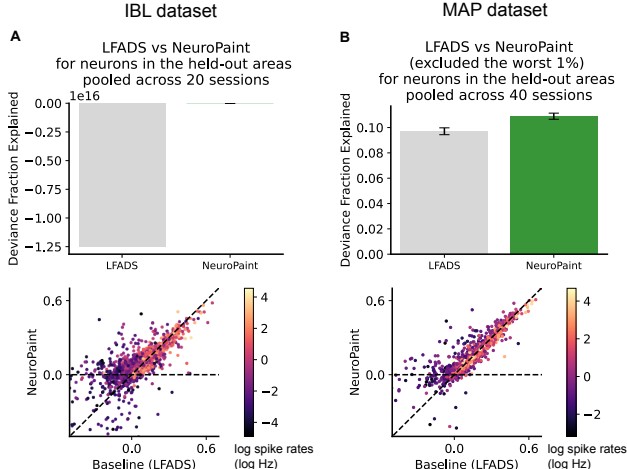

Supplemental Figure 1: Prediction performance for neurons from held-out brain areas across the two datasets. **(A)** Top: mean and standard error of DFE on the IBL dataset across all neurons, here LFADS yields a mean DFE on the order of $-1e16$, primarily due to a subset of neurons with extremely poor predictions; compare Fig. 3C where the 1% of neurons with the poorest predictions were excluded. Bottom: per-neuron DFE for all the held-out areas in 20 IBL sessions. **(B)** Top: mean and standard error of DFE on the MAP dataset, excluding the 1% of neurons with the poorest predictions; compare Fig. 3G, which included all neurons. Bottom: per-neuron DFE for all the held-out areas in 40 MAP sessions.

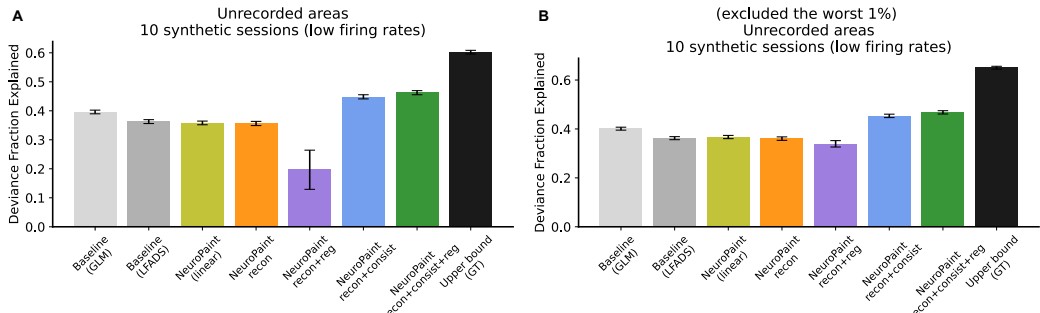

Supplemental Figure 2: Prediction performance from baselines, variants of NeuroPaint and ground truth firing rates for neurons from unrecorded areas in the synthetic dataset with lower firing rates. **(A)** Mean and standard error of DFE on the synthetic dataset with lower firing rates across all neurons. **(B)** Mean and standard error of DFE on the synthetic dataset with lower firing rates, excluding the 1% of neurons with the poorest predictions.

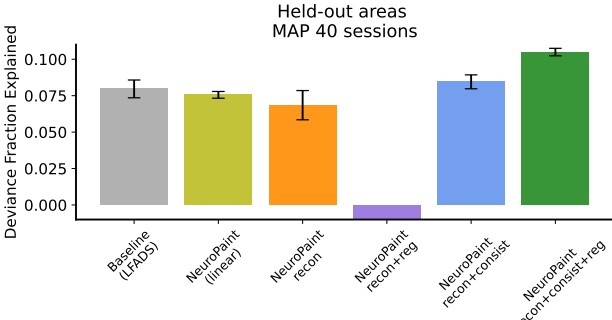

Supplemental Figure 3: Prediction performance from LFADS and variants of NeuroPaint for neurons from held-out brain areas in the MAP dataset. Bar plot shows mean and standard error of DFE on the MAP dataset across all neurons. For visuallization purpose, we truncate the y axis below –0.01. For NeuroPaint trained with reconstruction and regularization loss, its mean of DFE is -3.83, standard error of DFE is 0.11.

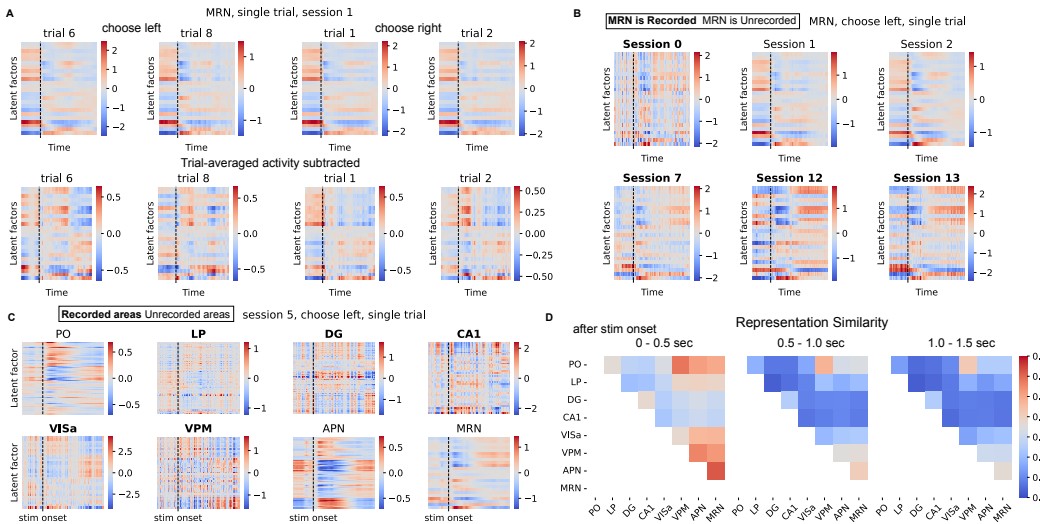

Supplemental Figure 4: *NeuroPaint's area-specific latent factors for the IBL dataset.* **(A)** Inferred latent factors for the Midbrain Reticular Nucleus (MRN), an unrecorded area in session 1 from the IBL dataset, during example trials across the *choose left* and *choose right* conditions. The bottom row shows the same latent factors with the trial-averaged activity subtracted. **(B)** Inferred single-trial latent factors for MRN (both recorded and unrecorded) across six sessions. **(C)** Latent factors for all eight areas (both recorded and unrecorded) in a *choose left* trial from a single session in the IBL dataset. **(D)** Representation similarity analysis (RSA) of latent factors across eight brain areas (recorded and unrecorded) for choose-left trials for three periods after stimulus onset in the IBL dataset, with values averaged across trials and pooled over sessions.

