# OpenReview forum: "Inpainting the Neural Picture: Inferring Unrecorded Brain Area Dynamics from  Multi-Animal Datasets"
_NeurIPS.cc/2025/Conference — NeurIPS 2025 poster_

### Official Review · Reviewer_Xqnt · 2025-06-25

**Clarity:** 4
**Significance:** 3
**Originality:** 4
**Rating:** 5
**Confidence:** 4

**Summary:**

This paper proposes a multi-session, multi-area masked Transformer model of large-scale neural activity with the stated goal of inferring unobserved neural activity from unrecorded brain areas. In general, through an architecture with area-specific components and training across sessions with variable recorded brain areas, the model infers a "brain wide" set of dynamics for all specified brain areas on each trial. With the addition of two losses that encourage capturing cross-area correlations and temporally smooth factors, the model outperforms a multi-session stitched LFADS model in heldout-neuron prediction. This evaluated was done on a simulated dataset as well as two larger-scale multi-area, multi-session neuropixel recordings. The authors show how the resulting model with area-specific factors enabled post-hoc analysis of neural representations.

**Questions:**

* Can the authors provide ablation results for NeuroPaint for the different losses on the IBL and MAP datasets? I understand these experiments are too expensive to run in this time window, but I'm wondering if they have been done and not reported. I think it will be important to report these results for a final version of this manuscript to better understand what components were important for achieving the best predictions. For example, does the model without the consistency and regularization losses also outperform LFADS?

* What is the relative weighting of the different loss terms? How was this chosen, and are the results sensitive to the relative weighting.

* Can the authors explore a higher-dimensional LFADS model with a larger encoder, controller, and generator, alongside higher dimensional latent factors that match the dimensions of NeuroPaint? My understanding is that the LFADS latent factors were 28, 11, or 27 which is significantly lower than NeuroPaints latent factor dimensions. Next, in related work I have seen LFADS models with 256 for modeling a single neuropixels session and a 512 dimensional generator for modeling stitched neuropixel sessions. I would encourage at least bumping up to 256 dimensions for the generator, controller, and encoder for the IBL and MAP datasets. The potentially limited capacity of LFADS here could explain why it does not adequately explain some neurons while providing strong predictions for others.

Overall, my evaluation score could increase with an improved reporting or discussion of how the model performs with and without the additional loss terms on the real datasets and/or by seeing that NeuroPaint also outperforms a larger LFADS model with ~256 dimensional RNNs and latent factors with larger dimensionality.

**Ethical Concerns:**

["NO or VERY MINOR ethics concerns only"]

**Final Justification:**

The authors' have addressed my questions and concerns, and I have raised my score by a point to reflect that. The paper will be improved with a greater emphasis on the importance of the consistency loss along with discussion of its potential limitations.

**Limitations:**

Yes

**Quality:**

3

**Strengths And Weaknesses:**

Strengths
* The proposed NeuroPaint architecture appears to be a very compelling approach to encode multi-area structure into the model, which addresses an important problem in the field. In particular, the ability to reason about putative latent factors across all brain areas on individual trials (shown in Fig. 4) is very nice.

* The evaluation across synthetic, IBL, and MAP datasets is strong and compelling. It is great to see the validation of the approach on the synthetic dataset in addition to testing on two difficult, large-scale neural datasets. In addition, the authors did a great job of clearly describing the model and providing many relevant details.

* The proposed approach provided improvement prediction on a difficult problem in two real neural datasets, which is an important contribution.

Limitations
* The consistency loss appears to have been an important part of the approach, enabling it to improve on LFADS in the synthetic dataset and potentially as well for the real datasets. However, this approach seems difficult to scale and compute, as it is computed for each pair of brain areas. It additionally makes a strong assumption that the correlation structure between each pair of areas should be fixed (within each trial type). The authors appear to allude to this limitation when discussing innovations needed to train across hundreds of brain areas.

* The language about what is being predicted about held-out areas could be improved. Initially, it sounds like spiking activity of heldout areas is being predicted unseen on new sessions; however importantly all models need to be aligned to a new heldout area on a new session with the GLM cross-validation. Instead, NeuroPaint provides a delineated set of latent factors for this new heldout area in a manner that the stitched LFADS architecture that mixes all regions together does not. For example, the final sentence on page 6 ("It is important...") should be altered, as it appears all models rely on post-hoc prediction. Instead, an important difference is NeuroPaint has separated the latent factors by region while LFADS has not.

* The proposed model requires a lot more compute for benefits that have not yet scaled with compute cost. For example, for the 40-session MAP models, LFADS requires 16 GPU hours (1 for 16 hours) whereas NeuroPaint requires 240 GPU hours (12 for 20 hours), so 15x compute cost compared to a DFE approvement of 1.25x for NeuroPaint.

* I understand it is time-consuming to tune LFADS. I appreciate the thorough comparison and reporting that the authors have already done. However, I expect the LFADS model is somewhat limited in capacity, and a larger model may close the gap (see questions).

---

> ### Author Rebuttal · Authors · 2025-07-30
>
> Thank you very much for recognizing the originality and strengths of our work. We address each point raised as follows.
>
> >The consistency loss appears to have been an important part of the approach, enabling it to improve on LFADS in the synthetic dataset and potentially as well for the real datasets. However, this approach seems difficult to scale and compute, as it is computed for each pair of brain areas. It additionally makes a strong assumption that the correlation structure between each pair of areas should be fixed (within each trial type). The authors appear to allude to this limitation when discussing innovations needed to train across hundreds of brain areas.
>
> We appreciate the reviewer’s insightful comments about the consistency loss. We would like to clarify a few important aspects regarding scalability and the assumptions involved.
>
> First, while the consistency loss has a computational cost that scales quadratically with the number of areas, we do not view this as a major limitation in practice. The total number of annotated areas in the mouse brain is on the order of hundreds. Importantly, even as the dataset grows larger in the future, the number of behaviorally relevant areas per experiment remains relatively limited. In contrast, the number of sessions in large-scale datasets can continue to grow – potentially reaching thousands in the future – so a method that scaled quadratically with session count would be far more problematic. Our design deliberately avoids such scaling, making it more suitable for future large-scale applications. We plan to explore alternative loss and model designs to reduce the computational cost, which we see as an important next step.
>
> Second, the consistency loss is applied not to the raw spike data or area-specific latent factors, but to the embedding factors (see Fig. 1C for definition), which are related to the neural activity through highly nonlinear transformations. In other words, the assumption about stable correlations between areas is imposed only at the level of these embeddings, making it less restrictive than it may appear. We will clarify this point in the revised manuscript.
>
> We agree that in more complex settings – such as neural recordings during trial-unaligned, naturalistic behaviors – the design of consistency loss might not be applicable. However, for trial-aligned data considered in our work, we believe the consistency constraint is reasonable and effective.
>
> >The language about what is being predicted about held-out areas could be improved ... For example, the final sentence on page 6 ("It is important...") should be altered, as it appears all models rely on post-hoc prediction. Instead, an important difference is NeuroPaint has separated the latent factors by region while LFADS has not.
>
> We agree with the reviewer that the language needs to be improved. We will revise the sentences to clarify that the important difference is that NeuroPaint has separate latent factors for each area, including the unrecorded ones.
>
> >The proposed model requires a lot more compute for benefits that have not yet scaled with compute cost. For example, for the 40-session MAP models, LFADS requires 16 GPU hours (1 for 16 hours) whereas NeuroPaint requires 240 GPU hours (12 for 20 hours), so 15x compute cost compared to a DFE improvement of 1.25x for NeuroPaint.
>
> We would like to emphasize that NeuroPaint’s contribution goes beyond improved reconstruction performance. A key advantage of NeuroPaint is its use of area-specific latent factors, which enables novel analysis of inter-area interactions and allow us to characterize how single-trial neural dynamics vary across brain areas (even for the unrecorded ones) during behavior. These capabilities are not readily supported by LFADS or similar methods.
>
> Furthermore, while the performance gain in DFE may appear modest, we note that LFADS is already a strong baseline, and improving upon it—especially across many sessions—is inherently challenging. In such cases, even incremental gains can reflect meaningful advances.
>
> To address the concerns about computational efficiency, in Appendix A.12, we also implemented a linear version of NeuroPaint. This variant achieves performance comparable to LFADS with much lower computational cost  (see Supplemental Figure 2, 3 for performances, and A.11 for its computational cost), offering a more lightweight alternative for scenarios where computational resources are limited. We agree that reducing the computational cost without sacrificing performance is an important direction for future work.
>
> >I understand it is time-consuming to tune LFADS. I appreciate the thorough comparison and reporting that the authors have already done. However, I expect the LFADS model is somewhat limited in capacity, and a larger model may close the gap (see questions)
>
> >Can the authors explore a higher-dimensional LFADS model with a larger encoder, controller, and generator, alongside higher dimensional latent factors that match the dimensions of NeuroPaint? ... I would encourage at least bumping up to 256 dimensions for the generator, controller, and encoder for the IBL and MAP datasets. The potentially limited capacity of LFADS here could explain why it does not adequately explain some neurons while providing strong predictions for others.
>
> We appreciate the reviewer’s thoughtful suggestion and agree that it is important to assess whether the performance gap is due to model capacity differences. To this end, we conducted additional experiments where we trained LFADS models on the MAP dataset using larger architectures: we increased the number of units in the encoder, controller, and generator to 256, and explored two latent dimensionalities—243 (matching the total number of NeuroPaint latent factors pooled across areas) and 49 (matching the maximum number of factors across areas in NeuroPaint). Additionally, to prevent the loss of LFADS from diverging to NaN, we reduced the learning rate from 3e-4 to 2e-4. With this adjustment, we confirmed that training remains stable and converged successfully under this lower learning rate.
> Here we report the deviance fraction explained (DFE, defined in A.7) metrics for neurons in the held-out areas pooled over 40 sessions.
>
> | Metric                                                                                   | LFADS (27 factors, used in Fig. 3G,H) | LFADS (256 units; 49 factors) | LFADS (256 units; 243 factors) | **NeuroPaint**      |
> |------------------------------------------------------------------------------------------|---------------------------------------|-------------------------------|-------------------------------|----------------------|
> | DFE (mean ± ste)                                                                         | 0.08 ± 0.006                          | -2.93e20                      | -9.12e20                      | **0.10 ± 0.003**     |
> | DFE (the 5% neurons with the poorest predictions were excluded; mean ± ste)                                              | 0.10 ± 0.002                          | 0.08 ± 0.002                  | -9.31 ± 1.31                  | **0.12 ± 0.002**     |
>
> In both cases, we found that NeuroPaint continued to outperform LFADS in terms of reconstruction accuracy. Interestingly, the higher-capacity LFADS models performed worse than the smaller model used in our main experiments, potentially due to overfitting given the increased number of parameters. These results suggest that the performance gap is not solely attributable to architectural capacity, and that the paradigm introduced in NeuroPaint is the key to its improved performance.
>
> >Can the authors provide ablation results for NeuroPaint for the different losses on the IBL and MAP datasets? ... does the model without the consistency and regularization losses also outperform LFADS?
>
> We thank the reviewer for highlighting the importance of understanding the contribution of different loss terms of our model. We have the ablation results for the MAP dataset in Appendix A.14 and Supplemental Figure 3, as well as for an additional synthetic dataset in Supplemental Figure 2. These analyses isolate the effects of the consistency and regularization losses. Our results show that both terms contribute to improved performance, with the consistency loss being particularly important—it is essential for NeuroPaint to outperform LFADS.
>
> >What is the relative weighting of the different loss terms? How was this chosen, and are the results sensitive to the relative weighting.
>
> We thank the reviewer for pointing out this missing detail. The relative weighting of the loss terms was selected to ensure that the reconstruction loss remains the primary driver during training, while the consistency and regularization terms serve as auxiliary constraints. Specifically:
>
> - The **reconstruction loss** is scaled by a weight of 1 and normalized by the number of timesteps and neurons.
> - The **consistency loss** is scaled by a weight of 1 and normalized by the number of area pairs.
> - The **regularization loss** is scaled by a weight of 0.1 and normalized by the number of timesteps, and latent factors.
>
> These scalings were chosen so that both the consistency and regularization losses remain at least an order of magnitude smaller than the reconstruction loss throughout training. We will clarify this point in the revised manuscript.
>
> We have not yet conducted a systematic sensitivity analysis on these weightings, and we agree this is an important direction for future work—especially to further improve robustness and generalization across diverse datasets.

---

> > ### Comment · Reviewer_Xqnt · 2025-08-03
> >
> > Thank you to the authors for your substantial and thorough response. I appreciate the pointer to Supp Fig 3 for the ablation results on the loss terms and the additional LFADS experiment. It is interesting that the additional capacity actually resulted in worse performance for the LFADS model. While I'm guessing different regularization hyperparameters could improve this, I believe the authors have put forth considerable and sufficient effort in this baseline already.
> >
> > The authors themselves state that the consistency loss is essential for NeuroPaint to outperforms LFADS, which is backed up by the results in SuppFig3 and the synthetic example. This leads me to believe the most important contributions are the separation into per-region latents and the consistency loss. However, I still have issues with the generalizability of the consistency loss. It requires delineating trial types to the model, which limits the ability of this approach to be applied generally to neural data at large scales that may have continuous variation in behavior, variation across animals, sessions, and trials, and simply behavior where discrete trial types are a priori unknown.

---

> > > ### Author Response · Authors · 2025-08-04
> > >
> > > We appreciate the recognition of our key contributions and thank the reviewer for the thoughtful follow-up comment on the consistency loss. We agree that the current design of consistency loss relies on delineated trial types, which will not be directly applicable to neural data with naturalistic behaviors. Nevertheless, we believe this approach can be extended to such settings. For example, one could segment the animal's behavior into syllables or infer behavioral latent states from neural/behavioral data, and then apply the consistency constraint to neural activity segments with similar behavioral syllables or states. We will acknowledge this limitation and include a discussion of possible extensions in the revision.

---

> > > > ### Author Response · Authors · 2025-08-06
> > > >
> > > > > Overall, my evaluation score could increase with an improved reporting or discussion of how the model performs with and without the additional loss terms on the real datasets and/or by seeing that NeuroPaint also outperforms a larger LFADS model with ~256 dimensional RNNs and latent factors with larger dimensionality.
> > > >
> > > > We provided the information you mentioned, is there more information that would be helpful?

---

> > > > > ### Comment · Reviewer_Xqnt · 2025-08-08
> > > > >
> > > > > Thank you to the authors for your responses and answers to my questions. I have no other questions, and I will take this response and discussion into account for my final review.

---

### Official Review · Reviewer_fqAe · 2025-06-29

**Clarity:** 2
**Significance:** 3
**Originality:** 3
**Rating:** 5
**Confidence:** 3

**Summary:**

Most neural recordings sample only a few brain areas. The authors propose a model named NeuroPaint to address this by treating each unrecorded area as a masked token and learning a low-dimensional latent trajectory for every area through a shared transformer. More specifically, a cross-attention read-in embeds spikes, the transformer captures cross-area interactions, and a GLM-style read-out plus smoothness and consistency losses rebuild the spiking activity. On synthetic RNN data and large Neuropixels mouse datasets, the model predicts spikes in held-out areas far better than GLM and LFADS baselines.

**Questions:**

See above.

**Ethical Concerns:**

["NO or VERY MINOR ethics concerns only"]

**Final Justification:**

This paper addresses a fascinating and fundamentally important problem in neuroscience: inferring the dynamics of unobserved brain areas. Several reviewers, including myself, requested additional results to support the proposed method. In their rebuttal, the authors successfully addressed most of our concerns by providing the requested additional analyses.

**Limitations:**

The authors discuss the limitations in Section 6, and I agree with their statements.

**Quality:**

3

**Strengths And Weaknesses:**

**Strengths**:

1. The model can predict spikes for brain areas that were not recorded by treating those areas as masked tokens, and it clearly outperforms both GLM and multi-session LFADS on synthetic data and on the IBL and MAP Neuropixels datasets.

2. A clear architecture of this model: cross-attention read-in, shared transformer, GLM read-out, with most weights shared across sessions helps the model generalize across animals and sessions.

**Weaknesses**:

1. Section 5.3 shows that the latent factors are smooth across trials and that their representational-similarity structure shifts between stimulus, delay, and response periods. These findings are still descriptive: correlations only tell us the factors repeat, and RSA only tells us areas become more similar during the response, without showing which task variables the latents encode. A straightforward suggestion is to add a decoding experiment. For each area, train a simple linear classifier to predict trial variables from that area’s latents, and report its accuracy alongside classifiers trained on LFADS or GLM latents of the same dimensionality. If the proposed latents truly capture context-dependent variability, they should outperform these baselines. The authors are free to design any quantitative decoding analysis they see fit, so long as it clarifies what information the latents encode and enables a direct comparison with competing models.

---

> ### Author Rebuttal · Authors · 2025-07-30
>
> We thank the reviewer for the overall positive comments and we would like to address the weakness as follows.
>
> >Section 5.3 shows that the latent factors are smooth across trials and that their representational-similarity structure shifts between stimulus, delay, and response periods. These findings are still descriptive: correlations only tell us the factors repeat, and RSA only tells us areas become more similar during the response, without showing which task variables the latents encode. A straightforward suggestion is to add a decoding experiment. For each area, train a simple linear classifier to predict trial variables from that area’s latents, and report its accuracy alongside classifiers trained on LFADS or GLM latents of the same dimensionality. If the proposed latents truly capture context-dependent variability, they should outperform these baselines. The authors are free to design any quantitative decoding analysis they see fit, so long as it clarifies what information the latents encode and enables a direct comparison with competing models.
>
> As also noted by Reviewer EKaW, we performed decoding analyses to assess what task variables are captured by the inferred latent factors. Specifically, we trained linear classifiers (logistic regression) to decode either stimulus (high vs. low auditory tone) or choice (lick left/right/no lick) from the latent factors of each held-out brain area. We found that the latent factors in most areas reliably support decoding for both stimulus and behavioral choice, demonstrating that they encode meaningful, task-relevant information.
>
> Stimulus was decoded using summed activity over the last 100 ms of the stimulus period, choice was decoded using summed activity over the last 100 ms of the delay period. Here we report the balanced accuracy (mean +/- std over sessions) as the decoding performance metrics, separately for each held-out area in the MAP dataset. The **bottom line** in each cell shows the balanced accuracy for decoding from inferred latent factors. As a comparison, we show the balanced accuracy for decoding from the held-out spike data in the **top line**. We found that decoding accuracy for inferred latent factors is typically slightly higher than that from held-out spike data, suggesting that the inferred latent factors successfully capture the stimulus/behavioral variability in the brain area. This decoding improvement also suggests that the inferred latent factors provide a denoised and more complete representation than the held-out population spike data subsampled from the area of interest.
>
> |      | ALM        | lOrb       | vlOrb      | Pallidum  | Striatum  | MRN       | SC        |
> |------------------------|------------|------------|------------|-----------|-----------|-----------|-----------|
> | Stimulus (chance level =0.5)  | 0.60±0.12  | 0.58±0.08  | 0.53±0.08  | 0.65±0.07 | 0.53±0.04 | 0.62±0.08 | 0.66±0.08 |
> |                        | 0.71±0.08  | 0.50±0.13  | 0.61±0.13  | 0.80±0.13 | 0.73±0.06 | 0.68±0.12 | 0.76±0.09 |
> | Choice (chance level =0.33)   | 0.40±0.15  | 0.41±0.13  | 0.37±0.12  | 0.46±0.17 | 0.58±0.13 | 0.45±0.09 | 0.52±0.21 |
> |                        | 0.50±0.14  | 0.51±0.13  | 0.57±0.14  | 0.68±0.12 | 0.63±0.11 | 0.63±0.16 | 0.70±0.16 |
>
> We would like to clarify that a direct comparison with LFADS and GLM latents is not feasible. LFADS does not produce area-specific latent factors, so decoding from LFADS latents cannot be fairly compared to decoding from NeuroPaint’s area-specific factors, especially considering that the decoding performance varies a lot across brain areas. The GLM baseline in our study is designed to predict spikes in a held-out area based on spikes in recorded areas; it does not involve latent factors, and therefore does not support a comparable decoding analysis.

---

> > ### Comment · Reviewer_fqAe · 2025-08-03
> >
> > Thank you for sharing the additional findings. Inferring the dynamics of unobserved brain areas is both interesting and important in computational neuroscience. I’ve raised my rating to 5.

---

### Official Review · Reviewer_Jqe4 · 2025-07-02

**Clarity:** 2
**Significance:** 2
**Originality:** 2
**Rating:** 3
**Confidence:** 4

**Summary:**

In this work, the authors introduce NeuroPaint, a approach for inferring the latent dynamics of unobserved brain areas. Specifically, the model is a transformer-based that is use brain-area masking to allow the model to predict the dynamics of the masked brain area given the  other observed brain area.

**Questions:**

All my concerns are listed above.

**Ethical Concerns:**

["NO or VERY MINOR ethics concerns only"]

**Limitations:**

yes

**Quality:**

2

**Strengths And Weaknesses:**

# Strengths

I love the motivation behind the approach and think this is an interesting and important direction for computation neuroscience.

# Weaknesses

I have a number of qualms with the paper. To start, I don't understand the claim that multi-area state space models can't infer the dynamics of unknown brain areas. This is a very straight forward extension of most (if not all) state-space models in both training and inference. Training a state-space model---for simplicity, I will focus on deep SSMs trained via the ELBO---would be exactly the same as the proposed procedure: the input to the encoder would be the observed brain areas but the decoder would still try and predict all of the brain areas. Inference would be straight forward as well: I would pass the observed brain areas into my encoder and my decoder would automatically predict all of the brain areas.

Next, there are a number of design choices that don't make sense to me. Firstly, the paper relies on the observation high-dimensional neural recordings can be explained by low-dimensional dynamics. Now, low and high dimensional are relative but in general the low dimensions should be significantly lower than the high dimensional neural recordings. But the authors don't seem to follow this. For the synthetic data, the authors state in lines 183-184 that each area contains 20-60 neurons per session but they choose the dimensionality of the latent factors for each dimension to be 24 (line 680 in the appendix). For the IBL dataset, most area and sessions have less than 100-150 neurons yet the dimensionality of the latent factors are quite large (84 for LP and 72 for DG, table 1 in the appendix); the dimensionality of the latent factors (relative to the observed number of neurons) is a little too large which seems to go against the author's hypothesis.

Lastly, I don't understand why after training NeuroPaint (which already has a linear Poisson decoder) the authors train another one! The original NeuroPaint Poisson decoder is a part of the model and should be used for decoding. The fact that a separate decoder needs to be trained afterwards makes me questions the usefulness of the approach.

---

> ### Author Rebuttal · Authors · 2025-07-30
>
> We thank the reviewer for pointing out several confusing parts of the paper, and we are willing to address these concerns as follows.
>
> >To start, I don't understand the claim that multi-area state space models can't infer the dynamics of unknown brain areas. This is a very straight forward extension of most (if not all) state-space models in both training and inference. Training a state-space model---for simplicity, I will focus on deep SSMs trained via the ELBO---would be exactly the same as the proposed procedure: the input to the encoder would be the observed brain areas but the decoder would still try and predict all of the brain areas. Inference would be straight forward as well: I would pass the observed brain areas into my encoder and my decoder would automatically predict all of the brain areas.
>
> We agree with the reviewer’s comment that, in principle, other architectures such as state-space models can be extended to infer neural activity in unobserved brain areas. However, we would like to clarify that our contribution is to explicitly design and evaluate a paradigm tailored for this setting. Our paradigm includes several key innovations: 1) we introduce area-specific latent factors for both recorded and unrecorded areas; 2) we employ a mask training strategy where both unrecorded and masked areas are represented by mask tokens; 3) we incorporate consistency loss to encourage our model to generalize to unrecorded areas.
>
> While it is certainly possible to implement a similar model using other backbones, such as deep SSM, we chose transformer layers due to their maturity and wide support, which made implementation and integration much more straightforward. As a proof of concept, we implemented a linear version of the NeuroPaint in Appendix A.12, where we replace the transformer layers with linear layers. And we showed that this variant of NeuroPaint can perform comparably with LFADS, indicating that the paradigm we designed is broadly applicable across architectural backbones.
>
> Extending NeuroPaint with a deep SSMs backbone would be an interesting future direction, as deep SSMs have shown potential in capturing long-range dependencies and enabling online inference.
>
> >Next, there are a number of design choices that don't make sense to me. Firstly, the paper relies on the observation high-dimensional neural recordings can be explained by low-dimensional dynamics. Now, low and high dimensional are relative but in general the low dimensions should be significantly lower than the high dimensional neural recordings. But the authors don't seem to follow this. For the synthetic data, the authors state in lines 183-184 that each area contains 20-60 neurons per session but they choose the dimensionality of the latent factors for each dimension to be 24 (line 680 in the appendix). For the IBL dataset, most area and sessions have less than 100-150 neurons yet the dimensionality of the latent factors are quite large (84 for LP and 72 for DG, table 1 in the appendix); the dimensionality of the latent factors (relative to the observed number of neurons) is a little too large which seems to go against the author's hypothesis.
>
> We appreciate the reviewer’s concern and the opportunity to clarify our design choice regarding the number of latent dimensions. While the number of latent factors may appear high relative to the number of recorded neurons in a given session, it is low compared to the total number of neurons in a brain area, which is often on the order of 100,000 in the mouse brain. The latent factors shared across sessions are intended to capture neural dynamics common to the entire area, rather than being tailored to each subset of recorded neurons in each session.
>
> While the ambient/embedding dimensionality of the latent factors may be moderately large (sometimes over 50 as the reviewer pointed out), their intrinsic dimensionality is often much lower. This is expected, as we do not impose orthogonality constraints on the latent factors. Empirically, we observe strong correlations among inferred latent dimensions (see Fig. 4C), suggesting low intrinsic dimensionality (see table below for ambient/embedding dimension versus intrinsic dimension for each brain area of interest in the MAP dataset). This observation is consistent with the core hypothesis that each brain area exhibits low-dimensional dynamics.
>
> We agree with the reviewer that the number of latent factors we chose may be higher than necessary. In future work, we plan to investigate how reducing the number of latent factors affects performance, with the goal of identifying the minimal dimensionality required to achieve optimal results.
>
> In the table below, we report the intrinsic dimension of the reconstructed neural activity by the inferred latent factors for each recorded area in the MAP dataset. Here we evaluated the intrinsic dimension by calculating the **participation ratio** of the predicted firing rates based on the latent factors for each area separately, where the participation ratio reflects the effective number of linear dimensions contributing to the variance.
>
> |                | ALM         | lOrb        | vlOrb       | Pallidum    | Striatum    | VAL-VM      | MRN         | SC          |
> |----------------|-------------|-------------|-------------|-------------|-------------|-------------|-------------|-------------|
> | **Ambient/embedding dimension** (number of latent factors) | 29          | 31          | 37          | 23          | 31          | 49          | 22          | 21          |
> | **Intrinsic dimension** (participation ratio, i.e. number of effective dimension; mean +/- std over sessions) | 4.71 ± 1.45 | 4.82 ± 1.40 | 4.21 ± 1.43 | 3.51 ± 1.36 | 4.19 ± 1.53 | 5.63 ± 1.40 | 3.90 ± 0.98 | 4.50 ± 1.05 |
>
> >Lastly, I don't understand why after training NeuroPaint (which already has a linear Poisson decoder) the authors train another one! The original NeuroPaint Poisson decoder is a part of the model and should be used for decoding. The fact that a separate decoder needs to be trained afterwards makes me questions the usefulness of the approach.
>
> We appreciate the reviewer’s question and the chance to clarify this point. In the trained NeuroPaint model, we learn a separate linear Poisson decoder for each recorded brain area in each session. However, for unrecorded areas, by definition, we do not have spike data and therefore we can not train decoders for them. To approximate performance on unrecorded areas, we hold out certain areas during training, train NeuroPaint on the remaining data, and then test whether the inferred latents for the held-out areas can support accurate activity reconstruction using a separately trained linear Poisson decoder. We will clarify this point in the revised manuscript.
>
> Importantly, our goal is not to infer neural activity in the unrecorded areas at single-neuron level, but to infer the neural dynamics at the level of latent factors. These area-specific inferred latent factors offer opportunities to address key neuroscience questions, such as how neural dynamics differ across areas and how brain areas interact during behavior.

---

> > ### Author Response · Authors · 2025-08-06
> >
> > We hope our rebuttal was helpful. If there’s anything else we can clarify, we would be happy to follow up.

---

### Official Review · Reviewer_EKaW · 2025-07-03

**Clarity:** 3
**Significance:** 3
**Originality:** 3
**Rating:** 4
**Confidence:** 4

**Summary:**

Monitoring and understanding neural population activity across brain areas is critical for the understanding of neural computation. However, in practice recordings fail to monitor all brain regions simultaneously. To overcome this limitation, this work introduces a new masking autoencoder approach to learn representation across animals and brain areas to predict neural activity in no recorded areas. The masking training strategy in combination with a reconstruction and consistency loss function allows for the generalization properties of the model. The model was tested in synthetic and two neural datasets outperforming alternative existing methods.

**Questions:**

Understanding data need would further illustrate the applicability of the model for experimental use. For example, what are the data demands to retrain the model from scratch? And using some transfer learning strategy? Would it be possible to train a foundation model of sorts that generalizes across regions, animals and/or species?

Does the model generalize to other recording methods like calcium recordings?

Regarding information content. How many latent dimensions were used for each brain region? Is that enough to capture intrinsic, stimulus and behavioral variability? Does one have to manually adjust this hyperparameter if not adequate?

**Ethical Concerns:**

["NO or VERY MINOR ethics concerns only"]

**Limitations:**

The authors mention some limitations, but clearly stating data demands is critical to also understand the impact and applicability of the method. Moreover, understanding the information content of the latent and reconstructed signals should be fully characterized to understand the limitations.

**Paper Formatting Concerns:**

No concerns.

**Quality:**

3

**Strengths And Weaknesses:**

The model addresses a critical need for neuroscience discovery and was clearly formulated. The training paradigm also shows innovation needed to address this problem. The model was evaluated in simulation and two neural recordings showing the promise of the approach.

While the model was evaluated in terms or reconstruction, it would be important to understand the relevance of the method to also measure stimuli or behavioral decodability to understand the information content present in the reconstructed neural activity. It would also be important to test, possibly in simulation, the model requirements and limitations, such as data demands, trial repeats, number of latent dimensions or number of recorded cells per area.

---

> ### Author Rebuttal · Authors · 2025-07-30
>
> Thank you very much for recognizing the innovation and importance of our approach. We respond to the questions and concerns raised as follows.
>
> >Understanding data need would further illustrate the applicability of the model for experimental use. For example, what are the data demands to retrain the model from scratch?
>
> We agree that understanding data requirements is important for assessing the model’s experimental utility. In this work, we showed that the model performs well with 40 sessions for the MAP dataset and 20 sessions for the IBL dataset, but we have not systematically explored the minimal number of sessions or neurons required for effective training. However, we believe that the amount of data needed to train the model from scratch depends on several factors specific to each dataset, including the complexity of the behavioral task, the selection of brain areas of interest, and the animal species. For example, if the behavioral task is fairly complicated with the animal demonstrating highly diverse behaviors across trials, then the data demand would be larger compared to the case where the animals exhibit stereotypical behaviors. As a result, we believe this question is best addressed empirically in each use case, rather than by reporting a fixed threshold.
>
> >And using some transfer learning strategy? Would it be possible to train a foundation model of sorts that generalizes across regions, animals and/or species?
>
> Yes, we believe it is possible to train a foundation model using our proposed architecture. One promising direction would be to pretrain the model on a diverse set of datasets that cover similar brain areas and behavioral task conditions. This pretrained model could then be fine-tuned on new datasets with limited data, enabling generalization across animals, regions, or even species (e.g. generalize across non-human primates and humans or across mice and rats). While we have not yet explored this direction, our framework is well-suited for such an extension and we see it as an exciting avenue for future work.
>
> >Does the model generalize to other recording methods like calcium recordings?
>
> Yes, our method can be extended to calcium imaging data with minimal architectural changes. Specifically, the GLM-based stitcher can be replaced with a linear stitcher, and the reconstruction loss can be modified from a Poisson negative log-likelihood to mean squared error (MSE), which is more appropriate for continuous-valued calcium signals.
> That said, the core motivation for our method—integrating data across sessions with partially-overlapping recorded areas—is less common in calcium imaging datasets. In typical calcium imaging experiments, especially wide-field imaging, the same set of brain areas are usually recorded across sessions.
>
> Nonetheless, an interesting future direction would be to apply our approach to stitch together high temporal resolution, small field-of-view recordings with overlapping brain areas to infer wide-field activity with improved temporal resolution. This could help mitigate the limitations of wide-field calcium imaging, which often suffers from low temporal resolution due to technical constraints.
>
> >Regarding information content. How many latent dimensions were used for each brain region? Is that enough to capture intrinsic, stimulus and behavioral variability? Does one have to manually adjust this hyperparameter if not adequate?
>
> The number of latent dimensions used in NeuroPaint for each dataset is detailed in Appendix section A.5. Rather than setting this hyperparameter manually, we used a principled procedure based on the participation ratio of the spike data across sessions to determine the number of latent factors (see A.5). This procedure is designed to ensure that the latent space is large enough to capture the variability in the data.
>
> As shown by the strong reconstruction performance evaluated by deviance fraction explained (DFE; defined in A.7) in Figure 3C,G, the inferred latent dimensions are sufficient to capture intrinsic variability. To evaluate whether the latent factors capture the stimulus and behavioral variability, we performed decoding analysis on the MAP dataset. Specifically, we used logistic regression with cross-validation to decode either stimulus (high/low auditory tone) or choice (lick left/lick right/no lick) from the latent factors for each held-out area (**bottom line** in each cell of the table, mean+/-std over sessions), and we compared the **balanced accuracy** to that decoding from held-out spike data (**top line** in each cell of the table, mean+/-std over sessions). Stimulus was decoded using summed activity over the last 100 ms of the stimulus period; choice was decoded using summed activity over the last 100 ms of the delay period.
>
> |      | ALM        | lOrb       | vlOrb      | Pallidum  | Striatum  | MRN       | SC        |
> |------------------------|------------|------------|------------|-----------|-----------|-----------|-----------|
> | Stimulus (chance level =0.5)  | 0.60±0.12  | 0.58±0.08  | 0.53±0.08  | 0.65±0.07 | 0.53±0.04 | 0.62±0.08 | 0.66±0.08 |
> |                        | 0.71±0.08  | 0.50±0.13  | 0.61±0.13  | 0.80±0.13 | 0.73±0.06 | 0.68±0.12 | 0.76±0.09 |
> | Choice (chance level =0.33)   | 0.40±0.15  | 0.41±0.13  | 0.37±0.12  | 0.46±0.17 | 0.58±0.13 | 0.45±0.09 | 0.52±0.21 |
> |                        | 0.50±0.14  | 0.51±0.13  | 0.57±0.14  | 0.68±0.12 | 0.63±0.11 | 0.63±0.16 | 0.70±0.16 |
>
> Note that due to randomness, no data from VM-VAL was held out in the MAP dataset. As a result, we are unable to report results for this area.
>
> We found that decoding accuracy for inferred latent factors is typically slightly higher than that from held-out spike data, suggesting that the inferred latent factors successfully capture the stimulus/behavioral variability in the brain area. This decoding improvement also suggests that the inferred latent factors provide a denoised and more complete representation than the held-out population spike data subsampled from the area of interest.

---

> > ### Comment · Reviewer_EKaW · 2025-08-01
> >
> > I appreciate the authors the detail comments. This is work gives a relevant direction for the neuroscience community, so I will maintain my current score.

---

### Note · Authors · 2025-08-13

We thank the reviewers for their constructive feedback and for recognizing the novelty, clarity and significance of our work. NeuroPaint is, to our knowledge, the first approach capable of inferring **area-specific neural dynamics** for both recorded and unrecorded brain areas, enabling single-trial, multi-area analyses not possible with prior methods.

**Key discussion points and resolutions**:

- **Data requirements & generalization** (EKaW) – We clarified dataset sizes used, explained how requirements depend on task complexity, recorded areas, and species, and outlined pretraining strategies for foundation models. Minimal modifications for calcium imaging were also described.

- **Information content of latents** (EKaW, fqAe) – Decoding analyses for stimulus and choice on the MAP dataset showed inferred latents often outperformed spike data, indicating denoised, task-relevant representations.

- **Conceptual/design issues** (Jqe4) – We explained distinctions from extended state-space models, highlighted our innovations (area-specific latents, mask training adapted for unrecorded areas, consistency loss), justified latent dimensionality using intrinsic dimension estimates, and clarified the rationale for training separate decoders when evaluating held-out areas.

- **Compute cost & consistency loss & baselines** (Xqnt) – We emphasized that area-specific latents enable novel inter-area analyses; presented a linear variant of NeuroPaint with LFADS-comparable performance at lower cost; clarified how consistency loss is computed and its assumptions, and scalability; as suggested, we showed that larger-capacity LFADS models still underperformed, indicating the advantage stems from our designed paradigm rather than model size.

- **Ablations & loss weighting** (Xqnt) – Ablations confirmed that both consistency and regularization losses improve performance, with consistency loss essential for outperforming LFADS. We reported relative weights for different loss terms and noted future sensitivity studies.

We are pleased that reviewer fqAe raised their score, and that reviewer Xqnt noted the possibility of increasing theirs and acknowledged that we had addressed their concerns. While reviewer EKaW maintained their score, they indicated that they were largely satisfied. Although reviewer Jqe4 has not responded yet, we have addressed all the points and are happy to provide additional explanation in the revised manuscript to improve clarity.

---

### Decision · Program_Chairs · 2025-09-17

**Decision:**

Accept (poster)

**Comment:**

This paper presents a deep learning-based technique for inferring the activity of missing brain areas in recorded neural datasets by leveraging multi-animal datasets with overlapping recordings across brain regions. This addresses an important practical challenge in neuroscience where complete brain-wide recordings are often not feasible from single animals.

The weaknesses initially identified by most reviewers have been adequately addressed, as reflected in their updated final assessments. One reviewer did not engage further in the discussion process, preventing me from confirming whether their specific concerns were resolved from their perspective. However, my own review of the authors' rebuttals indicates that the major concerns raised were at least partially clarified and addressed.

Given the importance of the problem this work tackles and the positive evaluations from the majority of reviewers, I recommend acceptance for this paper.